# Improving crystallization and eruption age estimation using U-Th disequilibrium dating of young volcanic zircon

Zoe Moser[1], Marcel Guillong[1], Chetan Nathwani[1], Kurumi Iwahashi[1,2], Razvan-Gabriel Popa[1], and Olivier Bachmann[1]

[1]Institute of Geochemistry and Petrology, ETH, Zürich, Switzerland
[2]Geological Survey of Japan, AIST, Tsukuba, Japan

**Correspondence:** Zoe Moser (moserz@eaps.ethz.ch)

**Abstract.** Quantifying timescales and establishing robust eruption chronologies is critical for understanding the evolution and hazards of volcanic systems. U-Th disequilibrium dating on zircon is especially valuable for young and active systems ($< 300$ ka). However, there is no consensus on how to calculate U-Th crystallization ages. To address this, we applied an optimized LA-ICP-MS U-Th-Pb double-dating strategy that simultaneously retrieves U-Th and U-Pb ages from the same zircon
ablation volume. This dating routine increases confidence in crystallization ages across 150-300 ka, where the resolution of either dating technique alone is limited. We applied this strategy to the Kos Plateau Tuff, which spans this critical interval, and compared U-Th model age calculation approaches against the well-established U-Pb age calculations. U-Th model ages calculated using the two endmember approaches, either using a constant melt composition or a constant zircon-melt U/Th fractionation factor ($f_{U/Th}$), yield similar age spectra when well-estimated values are used. In this context, it is essential to
evaluate whether the measured groundmass glass or whole-rock composition truly reflects the zircon-forming melt. This can be assessed by comparison with the youngest isochron intercept on the secular equilibrium line, which provides an independent melt composition estimate. We also evaluated eruption age estimation methods using synthetic U-Th datasets, with increasing uncertainty toward older ages. Bayesian models, particularly those with uniform priors, consistently outperformed weighted mean methods in terms of accuracy and precision and are therefore recommended for eruption age estimates in volcanic U-Th
zircon datasets.

## 1 Introduction

Quantifying eruption frequencies and timescales associated with magmatic processes is a fundamental goal in volcanology, particularly for long-lived systems leading to caldera-forming eruptions. Reconstructing the eruptive history of such complex systems, with countless eruptions over tens of millennia, presents significant challenges. Establishing a relative stratigraphy can be difficult, as individual volcanic units may not physically overlap. Additionally, large volcanic systems often exhibit
very active geothermal systems at the surface, leading to widespread alteration zones (e.g., Yellowstone: Fournier, 1989; Torfajökull: Björke, 2010; Campi Flegrei: Piochi et al., 2021). Such alteration frequently degrades minerals that would otherwise record magmatic conditions, rendering them ineffective for reconstructing volcanic histories. Zircon, however, is exception-

ally resistant to alteration (Watson and Harrison, 1983). And as it incorporates uranium and almost no lead during magmatic
crystallization, it is a useful mineral for dating using the uranium decay systems (Watson and Harrison, 1983). Due to the slow
diffusion of elements such as Pb, Th, and U within its crystalline structure, even at magmatic temperatures (Lee et al., 1997),
zircon crystals record their crystallization age (Costa, 2008; Bachmann, 2010), providing valuable insights into the duration
of magmatic processes, as zircon crystallization can span prolonged periods. Since accurate chronology is essential for un-
derstanding volcanic system evolution, zircon geochronology provides a critical foundation for investigating these dynamic
systems.

The most widely used geochronological techniques for zircon are U-Pb dating, applicable to crystals older than ∼100 ka
(Sakata, 2018), and $^{238}$U-$^{230}$Th disequilibrium dating (hereafter simply U-Th), which is suitable for ages younger than ∼380
ka (Schmitt, 2011). U-Th disequilibrium dating has gained prominence in recent decades (Reid et al., 1997; Coombs and
Vazquez, 2014; Locher et al., 2025), yet a debated aspect of U-Th disequilibrium dating of zircon is how to best determine
crystallization ages. To calculate these ages, each zircon analysis, reported as ratios of isotopic activities expressed using
parantheses ($^{230}$Th)/($^{232}$Th) and ($^{238}$U)/($^{232}$Th) (Reid et al., 1997), needs to be linked to the melt from which it crystallized to
form a two-point zircon-melt isochron. As each zircon model age depends on the melt composition, it is important to estimate
this value carefully. Two contrasting ideas dominate as to how the U-Th zircon-melt pair can be approximated (Fig. 1): One
idea assumes that a constant melt composition in terms of ($^{230}$Th)/($^{232}$Th) and ($^{238}$U)/($^{232}$Th), hereafter referred to as the
melt anchor point, can be linked to the individual zircon zircon activity ratios (Schmitt, 2011). The other idea suggests constant
fractionation of U and Th between zircon and melt (Boehnke et al., 2016), where the melt signature is estimated from the zircon
composition itself through a constant U/Th fractionation factor ($f_{U/Th}$, Eq. (1)). Choosing between these two approaches can
have implications for interpreting crystallization histories of volcanic samples.

As zircon crystals from single volcanic eruptions typically exhibit dispersed U-Th or U-Pb dates (over kyrs), protracted
crystallization in the underlying magmatic system is dated and not the eruption itself. To directly date the eruption, alternative
chronometers such as zircon (U-Th)/He thermochronology, $^{40}$Ar/$^{39}$Ar dating or radiocarbon dating can be employed. (1)
Zircon (U-Th)/He thermochronology records the time of cooling below the closure temperature of He of ∼200°C (Reiners
et al., 2002), which often corresponds closely to the eruption age (e.g., Friedrichs et al., 2020). This dating technique comes
however with many challenges, including the need for corrections related to initial $^{230}$Th-$^{238}$U disequilibrium, alpha-ejection
effects, pre-eruptive residence time, and accounting for internal age and compositional heterogeneity (Friedrichs et al., 2020).
(2) $^{40}$Ar/$^{39}$Ar dating is widely used on potassium-rich minerals or volcanic glass (e.g., Smith et al., 1996; Groen and Storey,
2022; Castellanos Melendez et al., 2023). This technique typically records the time when argon becomes trapped within the
mineral or glass, which is commonly assumed to coincide with eruption (Kelley, 2002a). However, the presence of excess
or inherited argon, as well as post-eruption alteration, can bias the resulting ages, complicating their interpretation (Kelley,
2002b; Ellis et al., 2017). (3) Radiocarbon dating of organic material buried or killed by volcanic deposits provides another
means of constraining the timing of eruption (e.g., Friedrich et al., 2006; Danišík et al., 2020). Its applicability is limited to
ages younger than ∼55 ka (Hajdas et al., 2021), and it requires the presence of suitable organic horizons directly beneath
volcanic deposits, conditions that may be difficult to meet when attempting extensive and comparable dating across a volcanic

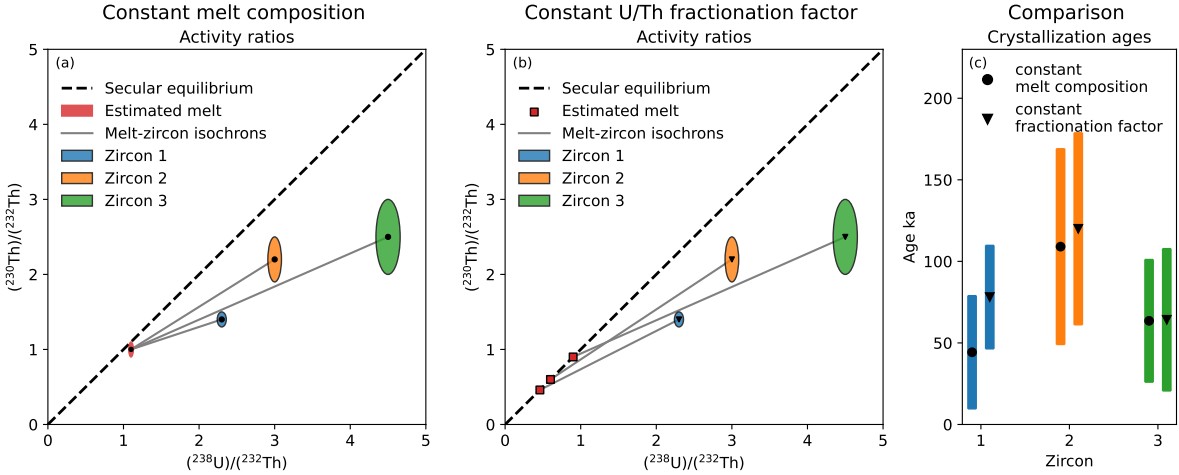

**Figure 1.** Illustration of two approaches for calculating model $^{238}$U-$^{230}$Th disequilibrium crystallization ages using three synthetic zircon crystals with distinct isotopic signatures (uncertainties represent $2\sigma$). (a) In the first approach, a constant melt composition (melt anchor point) is assumed to construct individual zircon-melt two-point isochron slopes (Schmitt, 2011). (b) In the second approach, a fixed U/Th fractionation factor ($f_{U/Th}$, Eq. (1)) between zircon and melt (here $5 \pm 1$) is applied to derive the model slope (Boehnke et al., 2016). (c) Resulting crystallization ages from both approaches are compared for identical zircon isotopic signatures.

system. Alternatively, we can infer the eruption age from the latest zircon crystallization, if we assume that zircon crystallized until the eruption took place (Keller et al., 2018; Nathwani et al., 2025). Protracted zircon crystallization, often reflected in a broad spread of non-overlapping dates within a single sample (Bachmann et al., 2007b; Schaltegger et al., 2015; Klein and Eddy, 2024), renders weighted mean and isochron ages derived from all analyses unreliable for eruption age estimation. In contrast, the youngest zircon age generally lacks statistical robustness (Keller et al., 2018). Consequently, alternative methods have been applied to estimate the eruption timing, including a weighted mean of a subset of zircon ages (e.g., Schoene et al., 2015; Locher et al., 2025) and likelihood-based Bayesian methods (e.g., Baudry et al., 2024; Cisneros de León et al., 2025). Keller et al. (2018) pioneered the application of Bayesian statistics for eruption age estimation from a distribution of zircon U-Pb ages. They found that this method produces superior accuracy and estimates of uncertainty relative to the weighted mean or youngest zircon methods. However, their study focused on ID-TIMS U-Pb data using synthetic datasets in which all ages were assigned the same uncertainty, a scenario more applicable to U-Pb than to U-Th datasets. In U-Th disequilibrium dating, uncertainties increase for older ages due to the exponential convergence toward secular equilibrium, making the zircon-melt isochron age increasingly sensitive to analytical uncertainties in the activity ratios (Schmitt, 2011). Despite the growing number of studies using U-Th zircon data, a range of eruption age estimation methods are currently employed without a systematic evaluation of which method yields the most reliable results.

To improve the application of U-Th zircon dates for establishing a fundamental part of the geochronological framework, and thereby contributing to quantifying eruption frequencies and magmatic timescales, we first evaluate the two opposing model

age approaches (constant melt vs. constant U/Th fractionation), followed by an assessment of different eruption age estimation methods (weighted mean vs. Bayesian) for typical U-Th zircon datasets measured by laser ablation inductively coupled plasma mass spectrometry (LA-ICP-MS). We test the consistency between the U-Th model age approaches and U-Pb ages, using a LA-ICP-MS strategy that simultaneously measures U, Th, and Pb isotopes in zircons from the Kos Plateau Tuff (KPT), which exhibit crystallization ages ranging from $\sim$160 to > 300 ka (Bachmann et al., 2007a; Guillong et al., 2014). Our applied U-Th-Pb LA-ICP-MS routine follows the general idea of Ito (2014, 2024), but differs in the optimization of dwell times to improve precision on minor isotopes (e.g., $^{206}$Pb, $^{207}$Pb, $^{230}$Th) and in the selection of measured masses to allow direct mass bias correction by including $^{235}$U while avoiding measurements of masses not required for our correction scheme (202, 204, 208). By comparing co-recorded $^{207}$Pb-corrected $^{206}$Pb/$^{238}$U (hereafter simply U-Pb) and U-Th ages from the same ablation volume, we assess the reliability of different ways to calculate U-Th model ages. To further evaluate eruption age estimation methods, we apply them to synthetic U-Th age datasets that simulate typical LA-ICP-MS uncertainties (Fig. 2). In addition, we date three samples with the classic U-Th method with independently well-constrained eruption ages, which serve as benchmarks to validate both the model age approaches and the eruption age estimation methods.

## 2 Methods

### 2.1 U-Th(-Pb) dating

#### 2.1.1 Samples

To assess the different approaches of calculating the U-Th crystallization ages (constant melt or constant $f_{U/Th}$), we developed an optimized LA-ICP-MS strategy to measure U-Th-Pb simultaneously (similar to Ito, 2014, 2024), and to compare the differently calculated U-Th ages with the U-Pb ages of the same ablation volume. In contrast to the approach by Ito (2014, 2024), who modified the U-Pb dating protocol to include Th measurements, we adapted the U-Th dating protocol of Guillong et al. (2016) to additionally measure Pb. This method was applied to zircon from the Kos Plateau Tuff, with prolonged zircon crystallization between roughly 160-300 ka (Bachmann et al., 2007a; Guillong et al., 2014). This age range is at the upper limit of potential U-Th age determination (Schmitt, 2011) and near the lower limit of $^{238}$U-$^{206}$Pb age resolution (Guillong et al., 2014). Although young U-Pb ages require corrections for initial U-Th disequilibrium and common lead, these corrections are well understood, making the overall age calculation relatively straightforward (Sakata et al., 2017; Pollard et al., 2023). In contrast, U-Th model ages depend on the isotopic composition of the melt in equilibrium with the zircon, for which there is currently no consensus on how to estimate it. Consequently, U-Pb ages provide a useful benchmark for evaluating the performance of different U-Th model age approaches (Sakata et al., 2017; Pollard et al., 2023).

We then validate these model age approaches by applying the best ones to U-Th datasets for samples with well-constrained eruption ages. For this purpose, we used the classical U-Th LA-ICP-MS measurement routine (Guillong et al., 2016). We chose to analyze one sample from the historic Heisei eruption of Mount Unzen in Japan between 1991-1995 and two samples, Laugahraun and Thórsmörk, from the Torfajökull volcanic system in Iceland (Moles et al., 2019). Laugahraun is a young lava,

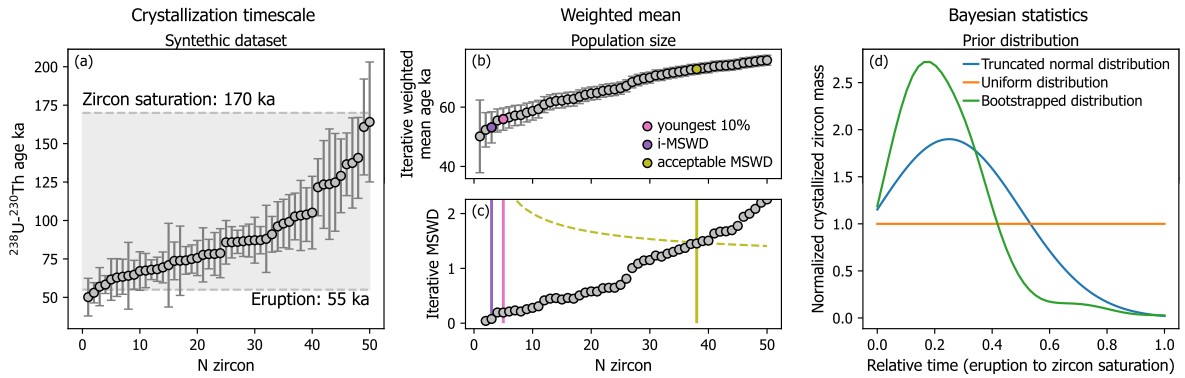

**Figure 2.** Example of a synthetic U-Th LA-ICP-MS zircon crystallization age dataset and comparison of eruption age estimation methods. (a) Synthetic ranked dataset of 50 zircon U-Th crystallization ages ($1\sigma$), simulating a natural volcanic system with ages skewed toward the eruption (Nathwani et al., 2025). Ages range from a preset eruption age of 55 ka to a zircon saturation age of 170 ka and include typical LA-ICP-MS uncertainties. (b) Iteratively calculated weighted mean ages ($1\sigma$). The colors correspond to the tested subsets of zircon ages for the weighted mean method to estimate the eruption age. (c) Corresponding iterative MSWD values, shown alongside the tested subsets of zircon ages (solid lines) used to estimate the eruption age via the weighted mean method. The "youngest 10%" weighted mean is calculated from the youngest 10% of zircon ages. The "i-MSWD" weighted mean corresponds to a subset of young zircon ages before, according to Popa et al. (2020), a visual increase in the iteratively calculated MSWD indicates the addition of older ages. The "acceptable MSWD" weighted mean follows a criterion as a function of the number of datapoints (N, dashed line) at which it remains possible that the zircon ages represent an isochron age (Wendt and Carl, 1991). (d) Relative zircon crystallization distributions between zircon saturation (1) and eruption (0). The distributions represent three different scenarios of normalized crystallized zircon mass as a function of time and are tested as prior distributions for the likelihood-based Bayesian eruption age model (Keller et al., 2018). While the truncated normal and uniform distributions are fixed, the bootstrapped distribution is obtained through the kernel density function of the zircon age dataset.

which erupted at $1480 \pm 11$ CE, determined through soil thickening rates and tephrochronology (Larsen, 1984). Thórsmörk is an ignimbrite that was most recently dated by $^{40}Ar/^{39}Ar$ measurements of glassy fiamme to an age of $56.14 \pm 0.44$ ka ($2\sigma$) by Groen and Storey (2022), which overlaps with other independently published ages (Svensson et al., 2008; Guillou et al., 2019; Moles et al., 2019).

The samples were prepared by crushing using a high-voltage selective fragmentation (SELFRAG) apparatus. Zircon crystals were then concentrated by heavy liquid separation with a sodium-polytungstate solution. Zircon crystals and groundmass glass shards were handpicked and mounted in epoxy resin. The grains were subsequently exposed by grinding and polished with a diamond suspension.

Additionally, we examined published SIMS U-Th zircon data from the Belford dome, samples SL-25 and SL-51, in the Soufrière Volcanic Complex in Saint Lucia (Schmitt et al., 2010; Barboni et al., 2016), in combination with whole rock data (Turner et al., 1996) that was used to calculate model ages. The eruption age of this dome was determined by (U-Th)/He

**Table 1.** LA-ICP-MS strategies for the three measurement routines applied in this study. The U-Th-Pb routine was used only for the KPT sample, whereas the typical U-Th routine was applied to the validation samples Heisei (Japan) and Laugahraun and Thórsmörk (Iceland). Groundmass glass was analyzed for all four samples. The main differences between the U-Th-Pb and U-Th routines are the measured masses, magnet positions, and dwell times, the latter influencing the precision and detectability of low-count isotopes. More technical details can be found in the supplementary Excel file.

| LA-ICP-MS strategy | U-Th-Pb zircon | U-Th zircon | U-Th groundmass glass |
|---|---|---|---|
| Measured samples | KPT (Greece) | Heisei (Japan) Laugahraun & Thórsmörk (Iceland) | KPT (Greece), Heisei (Japan), Laugahraun & Thórsmörk (Iceland) |
| Energy density | $\sim 2$ Jcm$^{-2}$ | $\sim 2$ Jcm$^{-2}$ | $\sim 3.5$ Jcm$^{-2}$ |
| Repetition rate | 5 Hz | 5 Hz | 5 Hz |
| Spot size | 29 $\mu$m | 29 $\mu$m | 163 $\mu$m |
| Ablation duration | 40 s | 40 s | 40 s |
| Background duration | 30 s | 30 s | 30 s |
| Masses measured | Pb206, Pb207, Th230, Th232, U235, U238 | 228[a], Th230, Th232, U235, U238 | Th230, Th232, U234, U235, U238 |
| Dwell times | 50 ms, 50 ms, 100 ms, 11 ms, 11 ms, 11 ms | 50 ms, 150 ms, 20 ms, 25 ms, 11 ms | 150 ms, 22 ms, 50 ms, 22 ms, 22 ms |
| Detection mode[b] | c,c,c,a,c,a | c,c,a,c,a | c,a,c,c,a |
| Magnet mass | 206, 207, 230, 230, 230, 230 | 228, 228, 228, 228, 228 | 230, 230, 230, 230, 230 |
| Primary reference materials | NIST612, 91500, Monazite, Zircon blank | NIST612, 91500, Monazite, Zircon blank | NIST612, Monazite |
| Validation reference materials | AUSZ7-1, GJ-1, Plesovice, FCT | AUSZ7-1, GJ-1, Plesovice, FCT | ATHO-G, BCR2G, BHVO2G |
| Data reduction scheme iolite | U-Pb reduction (young zircon) | U-Th reduction | U-Th reduction |

[a] $Zr_2O_3$: $^{90}Zr^{90}Zr^{16}O^{16}O^{16}O$

[b] c: counting, a: analog.

dating of zircon to $13.6 \pm 0.8$ ka ($2\sigma$) (Schmitt et al., 2010). Since these published data are used solely to validate the model age approaches, activity ratios obtained by either SIMS or LA-ICP-MS can be treated equivalently, because the analytical differences do not influence their use in the subsequent data treatment.

### 2.1.2 LA-ICP-MS measurement

The LA-ICP-MS measurements were conducted at ETH Zurich using a 193 nm Resonetics Resolution 155 LR excimer laser ablation system coupled to a Thermo Element XR sector field mass spectrometer. The specific parameters used for U-Th-Pb and U-Th measurements on zircon and U-Th measurements on the groundmass glass are summarised in Table 1. In all cases, masses with high expected countrates, such as $^{232}$Th and $^{238}$U, were measured with an analog detection mode and a common analog to pulse-counting equivalent factor (ACF) determined based on uranium before each session during calibration, while the other masses were measured in pulse-counting mode. For the U-Th-Pb LA-ICP-MS strategy, the magnet mass position was alternated between the low-mass Pb peaks (206, 207) and the higher-mass U and Th peaks ($\geq 230$), to minimize non-linear mass bias introduced by magnetic dispersion across the mass range. A repetition rate of 5 Hz, an energy density of $\sim 2$ Jcm$^{-2}$ with an ablation diameter of 29 $\mu$m for zircon and an energy density of $\sim 2.5$ Jcm$^{-2}$ with an ablation diameter of 163 $\mu$m for the groundmass glass was used. Reference materials such as NIST612 (Jochum et al., 2011), zircon 91500 (Wiedenbeck

et al., 1995), monazite, and a synthetic zircon, free of any U and Th (zircon blank), were measured alongside the unknowns. Additional zircons, AUSZ7-1 (Kennedy et al., 2014), GJ-1 (Jackson et al., 2004), Kara-18, Plesovice (Sláma et al., 2008),

FCT (Schmitz and Bowring, 2001), with known U-Pb ages and in secular equilibrium, were measured as secondary reference materials. For the groundmass glass measurement, ATHO-G, BCR2G, and BHVO2G were measured as secondary reference materials (Matthews et al., 2011). The data is available in the supplementary files.

All zircon measurements were conducted using a sample-standard bracketing protocol. Each analytical session began and ended with two consecutive analyses of the primary standards NIST612 and zircon 91500. These standards were repeatedly

measured in duplicate at regular intervals of 20-30 unknown analyses throughout each session. The zircon blank and secondary reference zircons were analyzed as single analyses at similar intervals but offset from the primary standards. The monazite was measured at the beginning, midpoint, and end of each session. At each time point, it was analyzed using three different ablation parameter sets (9 $\mu$m, 2 Hz; 9 $\mu$m, 3 Hz; and 13 $\mu$m, 3 Hz) to assess the abundance sensitivity of $^{232}$Th on $^{230}$Th with respect to different intensities, and to assess possible drift during a session. The protocol for the groundmass glass was

similar, except that only NIST612 was measured in duplicate, while the other validation reference glasses were analyzed as single measurement points.

### 2.1.3 U-Th data processing

Independent of the LA-ICP-MS measuring routine (U-Th-Pb or U-Th), the processing of the U-Th data follows the steps described by Guillong et al. (2016) and was done with a custom Data Reduction Scheme (DRS) written for implementation

in the Iolite software (provided as a supplementary file, also applicable for different minerals dated by U-Th disequilibrium). The DRS corrects the data for (1) the abundance sensitivity of $^{232}$Th on $^{230}$Th, (2) the interference of polyatomic zirconium oxide ($Zr_2O_3$) with mass 228 on mass 230, (3) the relative sensitivity between the measurement of U and Th, and (4) the mass bias (Guillong et al., 2016). To account for the zirconium oxide interference during the U-Th-Pb measurement without actively measuring mass 228, counts of mass 230 in a zircon blank were considered. With the measured secondary reference materials,

the secular equilibrium condition of ($^{230}$Th)/($^{238}$U) = 1 was evaluated. Throughout the sessions, a mean value of 1.002 (median of 0.994) was achieved, with a trend of higher variability and uncertainties for U-poor reference zircons (Fig. S1). Zircon Th and U concentrations were roughly estimated relative to the reference zircon 91500, with assumed values of 80 µg/g U and 30 µg/g Th (Wiedenbeck et al., 1995). The groundmass glass was processed similarly with the same DRS, but the correction for the interference of polyatomic zirconium oxide was not necessary.

With the processed data, we further calculated the individual zircon model ages with different approaches, which can be grouped into two main categories, both based on a two-point zircon-melt isochron approach (Fig. 1). (1) On the one hand, a constant isotopic melt composition can be assumed to have been in equilibrium with the individual zircon crystals (Schmitt, 2011). This melt composition can be approximated by either measuring the isotope ratios within the groundmass glass or in the whole rock, or by using an isochron intercept with the secular equilibrium line ($y_0$) as a melt anchor point. With this in mind,

we calculated the two-point zircon-melt isochron ages with different melt anchor point considerations: the equilibrium melt composition was approximated (a) through the measured ($^{238}$U)/($^{232}$Th) and ($^{230}$Th)/($^{232}$Th) activity ratios of the groundmass

glass including error propagation, (b) through the measured $(^{238}\text{U})/(^{232}\text{Th})$ activity ratio of the glass assumed to be in secular equilibrium (Boehnke et al., 2016) and equating the $(^{230}\text{Th})/(^{232}\text{Th})$ activity ratio to it, thus avoiding their high uncertainties, (c) through IsoplotR given the measured $(^{238}\text{U})/(^{232}\text{Th})$ activity ratio of the groundmass glass without a possibility of including a measurement uncertainty (Vermeesch, 2018) and (d) through the global (hereafter referring to calculations encompassing all analyses from the sample of interest) isochron intercept with the secular equilibrium line calculated by IsoplotR (Vermeesch, 2018). (2) On the other hand, a constant U/Th fractionation factor ($f_{U/Th}$) between zircon and melt can be assumed (Boehnke et al., 2016), where $f_{U/Th}$ is defined as the ratio of the uranium-to-thorium partition coefficients D between zircon and melt:

$$f_{U/Th} = \frac{D^U_{zircon-melt}}{D^{Th}_{zircon-melt}} = \frac{(U/Th)_{zircon}}{(U/Th)_{melt}}, \tag{1}$$

which corresponds to the ratio of the U/Th elemental ratios in zircon and the melt in equilibrium with the zircon (similar to Sakata et al., 2017). In this case, the fractionation factor needs to be approximated. Therefore, we additionally calculated the two-point zircon-melt isochron ages with different U/Th factionation factors between zircon and melt ($f_{U/Th}$) by using the model introduced by Boehnke et al. (2016): (a) $f_{U/Th} = 7$ as suggested by Boehnke et al. (2016), (b) $f_{U/Th} = 5$ as often this is the assumed value for initial $^{230}\text{Th}$ disequilibrium corrections (Guillong et al., 2014; Sakata et al., 2017), (c) $f_{U/Th} = 4$ as a reasonable estimate for a rhyolitic composition (Kirkland et al., 2015) and (d) $f_{U/Th}$ approximated through comparing the median zircon $(^{238}\text{U})/(^{232}\text{Th})$ with the measured groundmass glass.

The uncertainties on the measured $(^{238}\text{U})/(^{232}\text{Th})$ and $(^{230}\text{Th})/(^{232}\text{Th})$ activity ratios for zircon and groundmass glass, as well as the uncertainty on $f_{U/Th}$ (for which we used 0.8 at $2\sigma$ throughout the paper), were propagated according to the specific approach and tool applied. For the constant melt approaches (a) and (b), uncertainties were propagated using first-order Gaussian error propagation, whereas the other two constant melt approaches (c) and (d) followed the uncertainty propagation routines implemented in IsoplotR (Ludwig, 2003; Vermeesch, 2018). In the constant $f_{U/Th}$ approach, using the code by Boehnke et al. (2016), the uncertainties of the zircon and melt activity ratios, along with the preset uncertainty on $f_{U/Th}$, were propagated through a parametric bootstrap resampling method (Efron, 1992). The code additionally accounts for the potential spread of the modeled melt composition around the equiline, for which we applied the suggested value of 0.3 at $2\sigma$ (Boehnke et al., 2016).

Here, we point out that the original code by Boehnke et al. (2016) sets negative derived isochron-slopes throughout the resampling to zero, effectively interpreting them as zero ages. In young systems ($< 20$ ka), this procedure can artificially inflate model ages and bias statistical eruption age estimates. Therefore, to preserve unbiased results, we modified the code to retain negative slopes, and we recommend that others using this approach consider doing the same. While negative slopes are not geologically meaningful, retaining them preserves the statistical integrity and uncertainty structure of the resampling simulations.

For the estimation of the systematic uncertainties in U-Th measurements, we adapted the suggestions by Horstwood et al. (2016) for the U-Pb system to the U-Th system. (1, $s_y$) The U and Th concentrations of the 91500 reference zircon are unfortunately quite heterogeneous with standard deviations of 14% and 17% respectively (Jochum et al., 2005). This likely overestimates the true bias of the U/Th elemental ratio, as correction for relative U/Th sensitivity typically reproduces secular

equilibrium within 2% 2s. (2, $\varepsilon$') To our knowledge, there is no study of the long-term variance of a validation reference material for U-Th dating, which is why we assume 2% 2s. (3, $\lambda$) A systematic uncertainty of 0.15% 2s is further estimated from the [230]Th decay constant uncertainty (Cheng et al., 2013). (4, $\gamma$) A systematic uncertainty for the choice of model age approach is not considered, as it is part of this study to compare those approaches. However, even though the potential systematic bias for $\gamma$ is high (e.g. ~20%, Boehnke et al., 2016), it can be considerably reduced when well-constrained parameters are used. Quadratic addition of $s_y$ = 2%, $\varepsilon$' = 2% and $\lambda$ = 0.15% yields a total estimated systematic uncertainty of ~2.9% 2s. For the KPT U-Th model ages, specifically for comparison with the U-Pb ages, the systematic uncertainties were included through quadratic addition. Whereas for the other samples, the systematic uncertainties were applied only to the final eruption age results.

### 2.1.4 U-Pb data processing

For this study, U-Pb data were processed separately. Since we are dating young zircon (< 1 Ma), additional care is required. These crystals start with a U-Th disequilibrium at the time of crystallization due to U/Th fractionation. This disequilibrium is utilized when applying U-Th disequilibrium dating. However, to quantify the U-Pb age, it is necessary to correct for the initial [230]Th deficit to obtain accurate [238]U-[206]Pb ages (e.g., Schärer, 1984; Sakata et al., 2017). Moreover, because such young zircons have not accumulated sufficient radiogenic [207]Pb to overprint potential common lead contributions, greater emphasis must be placed on the common Pb correction and on the reliability of the [207]Pb/[206]Pb ratio (e.g., Sakata, 2018). Two primary methods exist for calculating isotope ratios from LA-ICP-MS data acquired during an ablation interval: the ratio of integrated intensities (ROI) and the mean of individual point-by-point ratios (MOR). They can diverge significantly when dealing with low-count isotopes such as [207]Pb and [206]Pb in young zircon (Ogliore et al., 2011). To mitigate the bias introduced by low-count statistics, especially in young and radiogenic Pb-poor zircon, we decided to adopt the ROI method to calculate the isotopic ratios. As the default U-Pb Geochronology DRS in Iolite calculates isotope ratios using the MOR method (Paton et al., 2010), we developed a custom DRS that first calculates ROI values, and then applies downhole fractionation, a relative sensitivity factor, and mass bias corrections to the ablation intervals using the corresponding time slices of primary reference zircon 91500 (Paton et al., 2010). Uncertainties were propagated using analytical Gaussian error propagation, combining the uncertainties of the isotope ratios and correction factors.

To calculate the U-Pb ages of the young volcanic zircon from the KPT sample to be later compared with the U-Th ages for the same ablation volumes, we used the DQPB application by Pollard et al. (2023) to retrieve the [207]Pb-corrected ages. The reported uncertainties of the ages follow the Monte Carlo uncertainty propagation (Sambridge et al., 2002). Initial disequilibrium was corrected by assuming a Th/U fractionation factor of $0.20 \pm 0.04$ ($1\sigma$) between zircon and melt, which lies between the published values of 0.25 and 0.18 for the KPT (Guillong et al., 2014). Using a Th/U fractionation factor of 0.25 to correct for initial Th disequilibrium would yield ages roughly 5 ky younger. To correct for common lead, an intercept of [207]Pb/[206]Pb = $0.8356 \pm 0.01$ ($1\sigma$) was used (Stacey and Kramers, 1975).

For the systematic uncertainty of the U-Pb system, we followed directly the suggestions by Horstwood et al. (2016) of (1, $s_y$) using 0.1% 2s uncertainty on the reference material as estimated from analytical precision of CA-ID-TIMS, (2, $\varepsilon$') the average

long term reproducibility of our laboratory with a conservative value of 1.2% 2s (Sliwinski et al., 2022), (3, λ) 0.08% 2s on the decay constants and $^{235}$U/$^{238}$U ratio (Jaffey et al., 1971; Cheng et al., 2013; Hiess et al., 2012), and (4, γ) for a systematic bias due to the commen Pb and Th/U fractionation correction an overall model systematic uncertatinty of 1.5% 2s is assumed. Quadratic addition of s$_y$ = 0.1%, ε' = 1.2%, λ = 0.08%, and γ = 1.5% yields a total estimated systematic uncertainty of ∼1.9% 2s. For the KPT U-Pb model ages, the systematic uncertainties were included through quadratic addition before comparing with the U-Th model ages.

To combine the U-Th and U-Pb ages for individual zircon analyses, a simple weighted average was used, with both analytical and systematic uncertainties considered in the weighting, since the two ages have distinct sources of systematic error. The systematic components from the individual U-Th and U-Pb ages, once propagated into the combined U-Th-Pb ages, are no longer systematic across the dataset, since each combined age is influenced by the specific weighting of the individual uncertainties. Therefore, we can not systematically reduce the uncertainty before applying the eruption age estimate. As a conservative compromise, we suggest additionally propagating the average systematic uncertainty of the two systems of 2.4%, quadratic addition of 2.9% and 1.9% to 3.5% seems unreasonable high, to the final eruption age estimates using the combined U-Th-Pb ages to avoid underestimating their uncertainties.

## 2.2 Model setup for eruption age calculation

### 2.2.1 Generating synthetic U-Th data

To evaluate the performance of different eruption age estimation methods, we generated synthetic U-Th age datasets with known eruption ages. We compared the ability of each method to reproduce the eruption ages most accurately and precisely. These synthetic distributions incorporate the analytical uncertainties ($\sigma$), number of zircons (N$_{zircon}$), and duration of zircon crystallization ($\Delta t$) relevant to natural U-Th LA-ICP-MS data.

(1) The model must randomly sample zircon dates from an underlying distribution representing zircon crystallization from zircon saturation until eruption. The choice of the underlying distribution is not straightforward, as the crystallization of zircon over time is not constant and depends on the individual temperature history and chemical composition of the systems (Schmitt et al., 2023). Magmatic zircon crystallization is well studied in terms of empirical saturation equations and kinetic models (Watson and Harrison, 1983; Watson, 1996; Boehnke et al., 2013), and predicts a peak in zircon crystallization at zircon saturation for monotonic cooling. Natural systems are more complex and do not strictly follow these simple temperature histories. Nathwani et al. (2025) demonstrated that zircon distributions from volcanic units tend to skew toward young ages, where crystallization of zircon has been truncated by the eruption, whereas plutonic zircon distributions reflect the kinetic models more accurately. Therefore, we decided to sample three different distributions to reflect the end members of zircon crystallization in volcanic systems: (i) a truncated normal distribution, (ii) a truncated monotonic cooling model used by Keller et al. (2017), and (iii) a uniform distribution. The uniform distribution is the simplest model that provides a geologically plausible distribution, suggesting relatively constant zircon crystallization throughout the period in which the system is zircon saturated. This can be translated into representing many non-resolvable periods of intensified zircon crystallization (Baudry

et al., 2024). The truncations in the other distributions represent the termination of crystallization as a result of the eruption, with the age peak skewing toward the eruption (Nathwani et al., 2025).

(2) After sampling the underlying zircon age distribution, typical uncertainties observed on U-Th zircon LA-ICP-MS datasets are assigned to the individual ages. Due to the inherent nature of the U-Th disequilibrium technique, progressively older zircon, as well as zircon with lower U/Th elemental ratios, will have higher uncertainties on their ages. While we can account for the age dependency by assigning higher uncertainties to the older ages, we also allow the uncertainty to spread around a mean value to account for variable U/Th elemental ratios. For illustration, this corresponds to $1\sigma$ uncertainties of approximately 4.5-

10.5 ka at 20 ka and 11-25 ka at 100 ka. Although the uncertainties on U-Th ages are not symmetric due to the non-linearity of the age equation (Ludwig, 2003), we apply symmetric uncertainties to the synthetic data for simplicity, as done in many publications (e.g. Baudry et al., 2024), model age tools (e.g. IsoplotR, Vermeesch, 2018), and eruption age estimation methods such as the Bayesian method (Keller et al., 2018) and weighted mean calculations. Finally, we add Gaussian noise to each age within a $2\sigma$ range of their assigned uncertainty to most accurately reproduce potential U-Th zircon age datasets.

For this study, we specifically simulated two different zircon crystallization periods: 0-40 ka (eruption at 0 ka, $\Delta t$ = 40 ky) and 55-170 ka (eruption at 55 ka, $\Delta t$ = 115 ky), to cover a range of lower and higher analytical uncertainties as well as different crystallization durations. For $N_{zircon}$, we choose values between 10 and 150, which are reasonable for typical LA-ICP-MS datasets (e.g. Jeong et al., 2024; Barboni and Bernal, 2025).

### 2.2.2    Eruption age calculation methods

The synthetically produced datasets were further used to reproduce the original preset eruption age underlying the sampled zircon ages using different methods. Two main families of methods were applied: the weighted mean and the likelihood-based Bayesian method. Both rely on assumptions, where the weighted mean requires a choice about the zircon age population included for the calculation, and the Bayesian method requires a prior distribution of the zircon ages (Fig. 2). While the weighted mean has the inherent assumption that all zircons within the chosen population have crystallized at the time of

eruption, the Bayesian method spells out the assumption of its prior knowledge in the form of the relative age distribution (Baudry et al., 2024). The two methods also propagate uncertainties differently. For the weighted mean, the uncertainties are calculated analytically using the standard formula for inverse-variance weighting (similar to Vermeesch, 2018). In the Bayesian method, uncertainties are derived from the distribution of eruption ages sampled via Markov Chain Monte Carlo (Metropolis et al., 1953). After discarding the initial burn-in samples, the mean and standard deviation of the remaining samples provide

the eruption age estimate and its $1\sigma$ uncertainty (Keller et al., 2018). Keller et al. (2018) has shown that for typical U-Pb age datasets, the performance of weighted mean methods differs depending on the degree of dispersion of the data, while the Bayesian method is less sensitive to this and is least likely to underestimate the uncertainties of its reported eruption estimate. Here, we test whether similar performance is observed for typical zircon U-Th LA-ICP-MS datasets.

    Using our synthetic zircon U-Th LA-ICP-MS datasets, we tested both methods. We applied the following weighted mean

methods to different sets of zircon age populations (Fig. 2b,c). (1) The first method is the "youngest 10%" weighted mean, where only the 10% youngest ages are included in the weighted mean calculation, with the purpose of keeping the proportion of

zircon constant while still having a statistically robust population. (2) The second method is the so-called "i-MSWD" weighted mean (Popa et al., 2020), where the mean squared weighted deviation (MSWD) is calculated iteratively, hence the name, by progressively adding older ages to the population. At each step, one additional older analysis is included, and the youngest coherent population is identified based on changes in the iterative MSWD values. These changes are evaluated visually from a plot, where distinct jumps in the MSWD indicate the inclusion of older age populations (Popa et al., 2020). (3) The third method is the "acceptable MSWD" weighted mean, where the population size is defined by the number of zircon at which the iteratively calculated MSWD gets closest to the accepted value, at which it remains possible that the zircon ages represent an isochron age (Wendt and Carl, 1991). For the Bayesian method, we choose to test three different prior distributions (Fig. 2d): (1) truncated normal distribution, (2) uniform distribution, and (3) bootstrapped distribution, where the kernel density function of the dataset itself serves as the prior distribution. Each method was evaluated in terms of its ability to reproduce the correct eruption age (accuracy) and its uncertainty (precision).

## 3 Results and discussion

### 3.1 U-Th model age calculation

#### 3.1.1 U-Th-Pb data of KPT zircon

The U-Pb ages from KPT predominantly span a crystallization timescale between 170-280 ka, with an average $2\sigma$ uncertainty of 20 ky per age. About 20% of the measured zircon had either common lead concentrations that were too elevated, or excessively high uncertainties on the $^{207}$Pb/$^{206}$Pb ratio, which prevented resolving a reliable crystallization age (Fig. 3). The U-Th ages based on the different approaches and approximations show a rough spread between 140-350 ka, with on average four times higher uncertainties compared to the U-Pb ages. For the constant melt approach, which uses the measured ($^{230}$Th)/($^{232}$Th) and ($^{238}$U)/($^{232}$Th) activity ratios in the groundmass glass, fewer model ages were resolved because many data points approached secular equilibrium and therefore produced infinitely large uncertainties. For the different U-Th model age approaches, individual U-Th and U-Pb ages are generally in good agreement within $2\sigma$ uncertainty (Fig. 4). However, IsoplotR U-Th model ages using the measured ($^{238}$U)/($^{232}$Th) activity ratio in the groundmass glass report lower uncertainties (on average half those of other U-Th ages), resulting in poorer agreement with the corresponding U-Pb ages compared to other U-Th model age approaches.

#### 3.1.2 U-Th model ages based on constant melt or constant $f_{U/Th}$

Overall, there is no significant difference in fit agreement between individual U-Pb ages and differently determined U-Th disequilibrium model ages (constant melt or constant $f_{U/Th}$), which is why we can not recommend one or the other approach conclusively (Fig. 4). Assuming a constant fractionation factor ($f_{U/Th}$) permits spatial and temporal heterogeneity in melt composition, which certainly happens in nature. Variations in U/Th elemental ratios of the zircon are then interpreted as originating from this heterogeneity (Boehnke et al., 2016). In contrast, a constant melt composition suggests that the magma

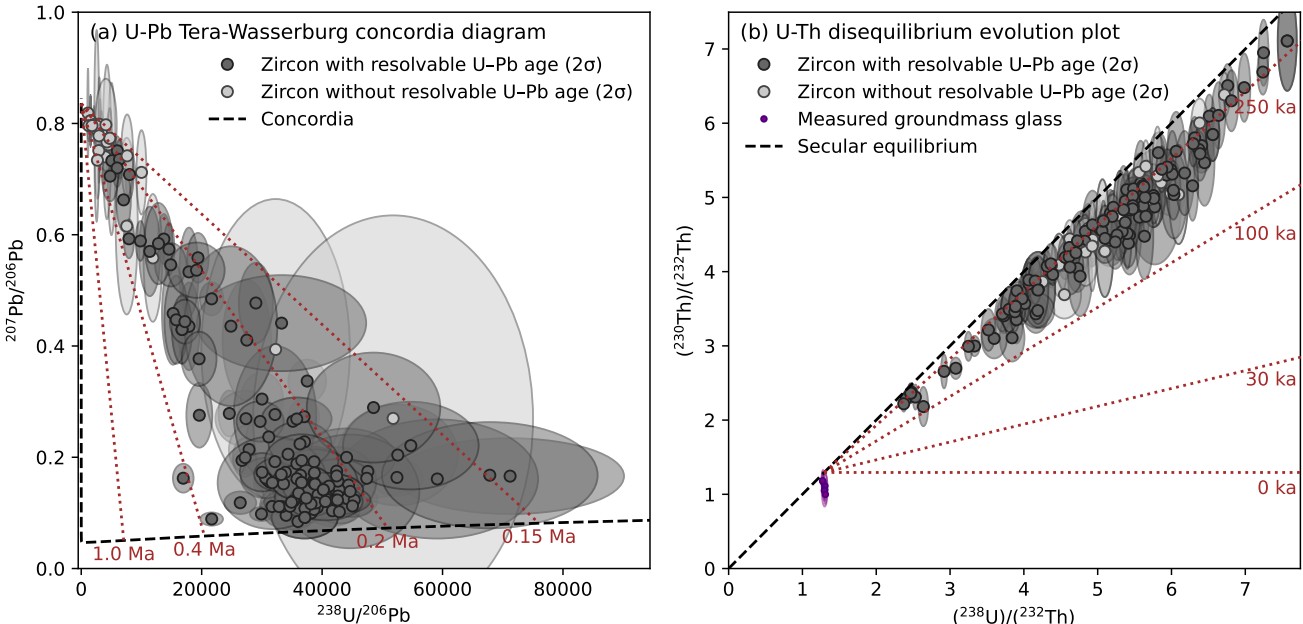

**Figure 3.** Overview of KPT data from U-Th-Pb double dating. All ellipses represent $2\sigma$ uncertainties. (a) U-Pb data are plotted on a Tera-Wasserburg diagram (Tera and Wasserburg, 1972). Measurements shown in light grey did not yield resolvable crystallization ages (Pollard et al., 2023), either due to elevated common Pb or high analytical uncertainties. The dashed black line represents the concordia calculated using $f_{\mathrm{Th/U}} = 0.2$ (Sakata, 2018), and red lines indicate reference age isochrons. (b) U-Th zircon and groundmass glass data are plotted on a U-Th disequilibrium evolution diagram. The dashed black line represents secular equilibrium, and red lines indicate reference age isochrons calculated using a melt anchor point at the intersection with the secular equilibrium line of the measured $(^{238}\mathrm{U})/(^{232}\mathrm{Th})$ activity ratio of the groundmass glass.

chamber was homogeneous during zircon crystallization, such that variations in the U/Th elemental ratio were negligible (Schmitt, 2011). Subsequently, any U/Th variations in the zircon would either result from variable U and Th partitioning

behavior or from mineral inclusions within the crystal, effectively mixing the zircon isotopic compositions with that of the inclusion. Trace element measurements reveal that many zircon crystals contain inclusions (Burnham, 2020), detectable via elevated La and P (apatite), Ti and Fe (Fe-Ti oxides), or Al and Fe (melt inclusions). Apatite inclusions notably affect the U/Th elemental ratio due to their Th affinity (Keller et al., 2022). However, since trace elements can not be measured in the same laser session as the U-Th disequilibrium dating, their presence can only be inferred indirectly through high $^{232}$Th counts

suggesting apatite inclusions. In such cases, assuming a fixed fractionation factor is incorrect, as these inclusions often have different partitioning behavior than zircon (Blundy and Wood, 2003), and is therefore more appropriately dealt with in the constant melt composition approach.

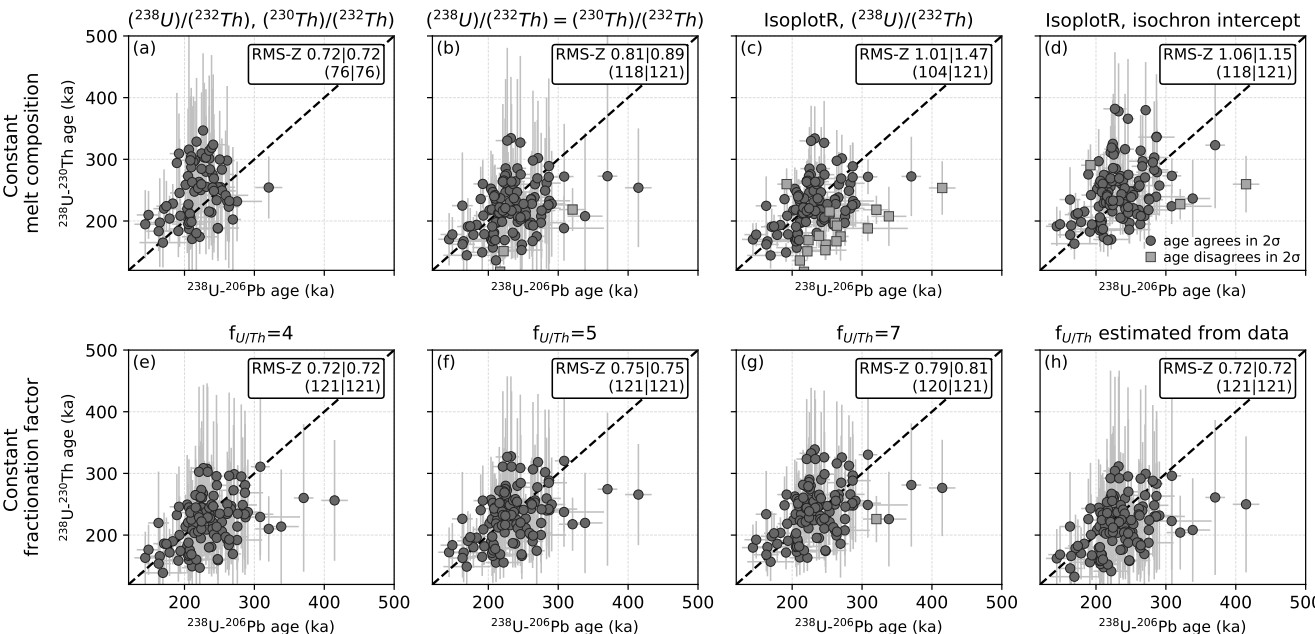

**Figure 4.** Comparison of KPT U-Pb and U-Th ages ($1\sigma$, including systematic uncertainties) from the same ablation volume using different U-Th age calculation approaches. Panels (a)-(d) show U-Th zircon crystallization ages calculated using a constant melt composition indicated by the title of each panel, while (e)-(h) use constant U/Th zircon-melt fractionation factors $f_{U/Th}$. Dark grey points represent data where U-Th and U-Pb ages agree within $2\sigma$; light grey squares indicate outliers with no overlap. Each panel shows the root mean square of the age differences normalized by the combined uncertainties of both datasets (RMS-Z) for the overlapping and total age datasets, along with the number of points considered (overlapping | total). RMS-Z values $\geq 1$ indicate decreasing agreement between the U-Th and U-Pb age estimates.

However, the constant melt approach may be questioned when many accessory phases crystallize in the boundary layers of growing minerals (Bacon, 1989; Bindeman and Melnik, 2016). In these regions, the melt can become locally enriched in incompatible elements such as U and Th, potentially fractionating them and altering the composition of the melt from which the zircon grows. Moreover, on a larger scale, magmatic systems are rarely homogeneous in space and time (Bachmann and Huber, 2016; Cashman et al., 2017). Melt can be influenced by pre-eruptive mixing processes (e.g. Nakamura, 1995; Troll et al., 2004), and zircon datasets show large variations in trace elements (e.g. Troch et al., 2018; Bell and Kirkpatrick, 2021; Ratschbacher et al., 2024), undermining the assumption of a constant melt composition.

Zircon crystallization is further complicated by undercooling and disequilibrium growth. Gillespie et al. (2025) showed that undercooling can lead to dendritic growth in zircon, where apparent cathodoluminescence oscillatory growth rims exhibit variable U/Th elemental ratios. During this process, the crystal corners grow rapidly and incorporate a lower U/Th elemental composition, whereas slower infilling into the planar structures produces higher U/Th values. This disequilibrium growth

explains U/Th elemental variability within zircon without invoking changes in melt composition or inclusions, highlighting that bulk partition coefficients may not accurately describe zircon grown out of equilibrium (Gillespie et al., 2025). Similarly, sector zoning represents a form of homogenous disequilibrium within zircon (Watson and Liang, 1995) and can influence trace element partitioning, even in a chemically homogeneous melt (Burnham and Berry, 2012; Burnham, 2020). Such zoning reflects variations in crystallographic orientation that can locally modify partition coefficients and result in heterogeneous trace element distributions within a single crystal (Burnham and Berry, 2012). Beyond crystal-scale effects, it has also been shown that $f_{U/Th}$ tends to decrease with increasing crystallization temperatures (Liang et al., 2025), while $f_{U/Th}$ was shown to decrease with increasing silica in the whole rock (Kirkland et al., 2015). Together, these factors highlight that U and Th partitioning is sensitive to both crystal-scale and system-scale processes, casting doubt on the assumption of a constant fractionation factor between zircon and melt. This underscores the complexity of the zircon-melt system and highlights that both model age approaches (constant melt versus constant $f_{U/Th}$) rely on end-member assumptions.

### 3.1.3   Estimating constant melt signature or constant $f_{U/Th}$ for obtaining model ages

Estimating the melt in equilibrium with the crystallizing zircon and defining a representative $f_{U/Th}$ is challenging. Using both measured melt ratios, $(^{230}Th)/(^{232}Th)$ and $(^{238}U)/(^{232}Th)$, in a zircon-melt isochron approach showed good agreement with U-Pb ages for the KPT zircon (Fig. 4). Yet, high $(^{230}Th)/(^{232}Th)$ uncertainties limit the number of resolvable ages. There is an observed scatter in glass $(^{230}Th)/(^{232}Th)$ ratios due to low counts, therefore limited precision, and minor systematic shifts can significantly influence zircon crystallization ages. Due to higher concentrations in the melt, measuring $(^{238}U)/(^{232}Th)$ in the groundmass glass can be done more accurately and with higher precision. Therefore, a more conservative approach assumes the melt is in or close to secular equilibrium, supported by Boehnke et al. (2016), using the measured $(^{238}U)/(^{232}Th)$ in the groundmass glass. This approach performed best under the constant melt composition models in comparison with the U-Pb ages and is therefore a preferred way of applying the constant melt anchor point. This is similar to the isoplotR calculation with a measured $(^{238}U)/(^{232}Th)$ activity ratio as a melt anchor point, but the measurement uncertainty is propagated. This error propagation has proven important, as many model ages calculated with isoplotR did not overlap with the U-Pb ages due to uncertainty underestimation (Fig. 4). For the KPT case, using a global isochron intercept performed less well relative to the U-Pb age benchmark than using the measured glass composition.

The constant fractionation models performed equally well compared to the constant melt approach for various fractionation approximations (Fig. 4). Among constant fractionation factors, the one using the median zircon compositions compared to the measured glass (for KPT: $f_{U/Th}$ ~3.6) performed well. Using $f_{U/Th}$ = 7, as suggested by Boehnke et al. (2016), did not perform as well as the others, most likely because the $f_{U/Th}$ is overestimated for a rhyolitic system like the KPT (Kirkland et al., 2015). However, given the close match between $f_{U/Th}$ = 4 and $f_{U/Th}$ = 5 models, and that U-Pb ages were corrected using a $f_{U/Th}$ = 5, this appears most suitable and consistent for U-Th age calculations.

Importantly, for older zircon, changing the constant melt composition, as well as a change in the constant fractionation factor, is less significant compared to young zircon with the same analytical uncertainty (Fig. S2). The reduced sensitivity of older zircon to variations in melt composition or fractionation factor may account for the lack of significant differences observed in

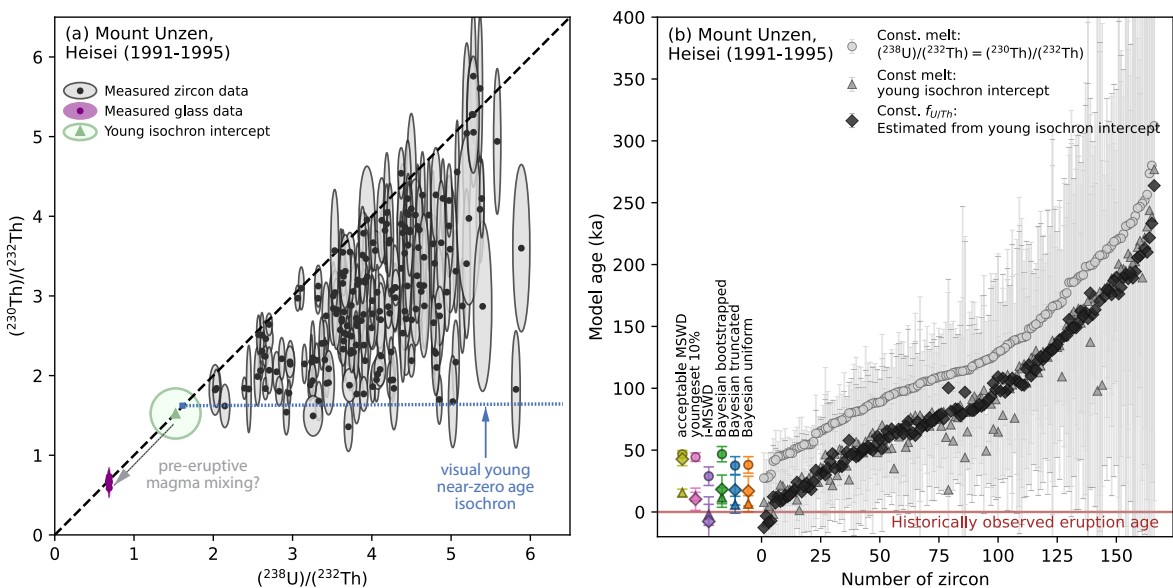

**Figure 5.** Validation of U-Th zircon age determinations using the Heisei sample from the historically observed 1991-1995 Unzen eruption (all uncertainties correspond to $2\sigma$, eruption estimates additionally include a systematic uncertainty of 2.9%). (a) U-Th evolution plot showing measured zircon and groundmass glass activity ratios, the young isochron intercept with the secular equilibrium line evaluated using IsoplotR (Vermeesch, 2018), and the visually identified young near-zero age isochron. The latter was drawn to ensure that the youngest age is within $2\sigma$ of the known near-zero eruption age. The discrepancy between the measured groundmass glass and the young isochron intercept indicates that the groundmass glass does not represent the melt from which the zircon crystallized. (b) Zircon crystallization ages using different approaches are plotted in grey tones, while eruption age estimates for the Heisei sample, calculated using different methods, are shown in color. The first three estimates represent weighted mean ages calculated from different subsets of zircon ages, whereas the remaining three are Bayesian eruption estimates based on different prior distributions. The marker shape corresponds to the different model age datasets. Measurements of enhanced Th, implying apatite inclusions were ablated along with the zircon, were only used to calculate model ages with the constant melt approach.

the KPT crystals when comparing model ages derived using different assumptions of constant melt composition or constant fractionation factor. Therefore, we applied different model age approaches to younger samples with well-defined eruption ages

to validate the findings made based on the KPT zircon.

### 3.1.4  Young isochron intercept to validate U-Th model ages

The Japanese sample, Heisei, from the historic eruption at Unzen volcano, illustrates that measured groundmass glass compositions may not reliably reflect the zircon-forming melt. The zircon crystals show prolonged crystallization, by plotting into a wedge-shaped field forming an almost perfect sphenochron (Fig. 5a). About 9% of the data points show strongly elevated

$^{232}$Th concentrations of $> 485$ µg/g Th, identified as statistical outliers using the 1.5 times interquartile range (IQR) criterion

(Tukey, 1977), indicating possible contributions from apatite inclusions. These measurements were therefore excluded from the model age calculations based on the constant $f_{U/Th}$ approach, as it assumes pure zircon-melt fractionation. No measurement falls below a $(^{230}Th)/(^{232}Th)$ activity ratio of $\sim$1.5, while in the groundmass glass activity ratios of $(^{238}U)/(^{232}Th) = 0.684 \pm 0.006$ (all uncertainties are reported at $2\sigma$ unless otherwise specified) and $(^{230}Th)/(^{232}Th) = 0.66 \pm 0.08$ have been measured.

Even the youngest calculated model age (27 $\pm$ 15 ka) using the measured groundmass glass either as a melt anchor point or to estimate the fractionation factor is significantly older than the eruption age at 1991-1995 (Fig. 5b). There is no evidence for zircon resorption that could explain the gap between the eruption age and the latest recorded zircon crystallization, nor is there any indication that zircon growth had ceased. Although it cannot be determined conclusively whether zircon continued to crystallize until the time of eruption, the evolution plot shows that the low-$(^{230}Th)/(^{232}Th)$ data form an almost horizontal array,

consistent with a young, near-zero-age isochron for this historic sample. However, this young isochron intercepts the secular equilibrium line significantly above the measured groundmass glass. This is a strong indication that, for this sample, the fresh and microlite-rich groundmass glass (Noguchi et al., 2008) is not representative of the melt from which the zircon crystallized. This eruption was strongly influenced by immediately pre- and syn-eruptive mixing between resident silicic, zircon-bearing crystal mush in the upper crust, and more mafic recharge coming from deeper in the system (Nakamura, 1995). Such mixing

likely remobilized zircon from the crystal mush, consistent with the observed long zircon crystallization timescales, and caused significant changes in melt composition, so that the erupted melt was no longer in equilibrium with any of the zircons from this sample. A possible solution to this issue is to estimate the melt composition through fitting a young isochron over the sphenochron and using its intercept with the secular equilibrium line ($y_0$) as a constant melt anchor point. To estimate $y_0$, we applied a workflow of calculating the isochron of the whole dataset using isoplotR (Vermeesch, 2018) and iteratively removing

the oldest ages and recalculating the isochron until the MSWD reaches a value of 1. From there, we retrieved $y_0$. For this Heisei sample, this approach suggested an intercept of 1.52 $\pm$ 0.32. By using this value as the constant melt composition to calculate the model ages, the crystallization timescale of this sample is more accurately represented, as the youngest crystallization ages overlap with the known eruption age. Estimating the $f_{U/Th}$ through the isochron intercept of 2.7 also yielded more appropriate crystallization ages. However, as this sample is strongly influenced by apatite inclusions, the assumption of a constant $f_{U/Th}$

becomes increasingly difficult to justify. Therefore, for this sample, we suggest calculating the crystallization ages through a constant melt approach by using the young isochron intercept as a melt anchor point.

Similarly, for the Belford dome (Schmitt et al., 2010; Barboni et al., 2016), using the whole rock composition, $(^{230}Th)/(^{232}Th) = 0.85 \pm 0.07$ and $(^{238}U)/(^{232}Th) = 0.72 \pm 0.12$ (Turner et al., 1996), as a constant melt anchor point overestimates its zircon crystallization ages (Boehnke et al., 2016), assuming no resorption took place and zircon crystallized until eruption (Schmitt

et al., 2010). Boehnke et al. (2016) has shown that calculating the crystallization ages using a constant $f_{U/Th}$ of 7 more accurately aligns with the proposed eruption age of 13.6 $\pm$ 0.8 ka (Schmitt, 2011). Using a young isochron intercept of 1.3 $\pm$ 0.4, which notably does not overlap with the measured whole rock signature (Fig. 6a), as a constant melt anchor point, results in very similar crystallization ages compared to using a constant $f_{U/Th}$ of 7 (Figure 6b,c).

The Icelandic samples, Laugahraun (LH) and Thórsmörk (TH), do not show very prolonged crystallization but rather re-

stricted sphenochrones (Fig. S3). The young isochron intercept at 1.05 $\pm$ 0.2 for LH and 1.18 $\pm$ 0.4 for TH overlap with their

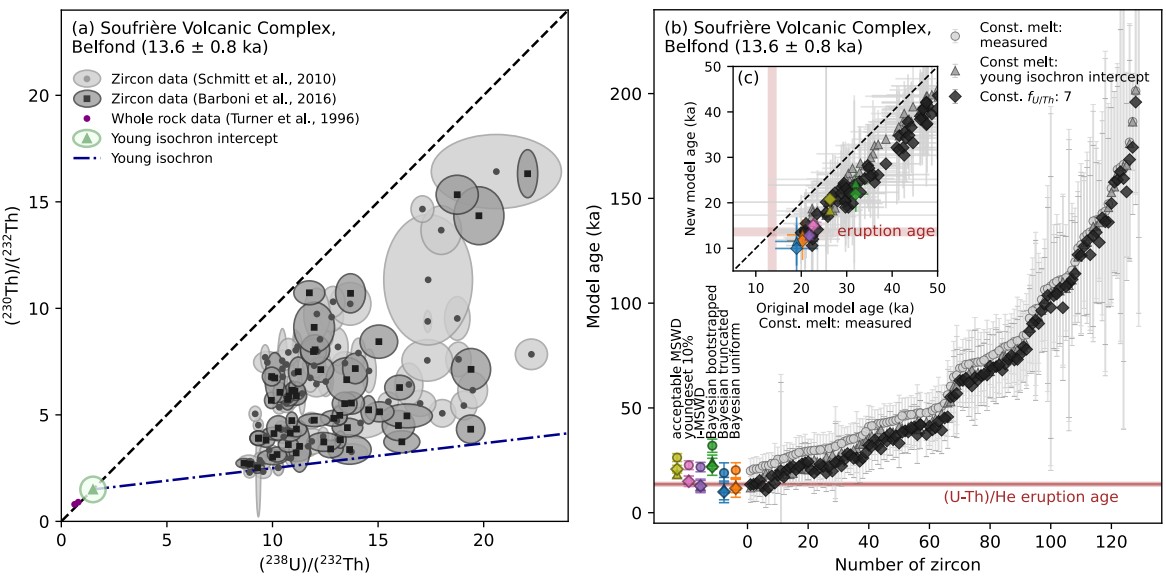

**Figure 6.** Literature-based zircon U-Th data from Belfond Dome (SL-25, SL-51) used as a second validation case study (all uncertainties correspond to $2\sigma$). (a) U-Th evolution plot showing published zircon activity ratios (Schmitt et al., 2010; Barboni et al., 2016), the whole-rock composition (Turner et al., 1996), a potential young isochron and its intercept with the secular equilibrium line at $y_0 = 1.3 \pm 0.4$. The discrepancy between the whole-rock composition and the young isochron intercept suggests that the whole rock is not representative of the melt from which the zircon crystallized. (b) Zircon crystallization ages calculated using three approaches (grey symbols) and eruption age estimates (colored symbols). The first three estimates represent weighted mean ages calculated from different subsets of zircon ages, whereas the remaining three are Bayesian eruption estimates based on different prior distributions. The Belfond sample was measured by SIMS. For simplicity, we have propagated similar systematic uncertainties as for the LA-ICP-MS U-Th data of 2.9% into the eruption estimates. The marker shape corresponds to the different model age datasets. A close-up of the youngest crystallization ages is shown in (c). The three model age datasets correspond to the originally published ages using the constant whole-rock melt composition (Schmitt et al., 2010), compared to a constant $f_{U/Th}$ of 7 (Boehnke et al., 2016), and a new constant melt approach of using the young isochron intercept. Eruption age estimates are compared to the (U-Th)/He eruption age of $13.8 \pm 0.8$ ka (Schmitt et al., 2010).

corresponding groundmass glass signatures: $(^{238}\text{U})/(^{232}\text{Th}) = 0.968 \pm 0.006$ and $0.97 \pm 0.06$, $(^{230}\text{Th})/(^{232}\text{Th}) = 1.03 \pm 0.12$ and $0.97 \pm 0.12$, for LH and TH respectively. This suggests that the melt from which the zircon grew is well represented by the groundmass glass. Apatite inclusions are evident in about 6% of the measurements based on enhanced Th concentrations (> 425 µg/g for LH, > 155 µg/g for TH) identified as statistical outliers. For those measurements, only the model ages calculated with the constant melt approach were calculated. The model ages that use the measured $(^{238}\text{U})/(^{232}\text{Th})$ in the groundmass glass and assume secular equilibrium are similar within uncertainty to those calculated using a constant $f_{U/Th}$, approximated by the median zircon and groundmass glass $(^{238}\text{U})/(^{232}\text{Th})$ activity ratios of 3.8 and 5.8 for LH and TH, respectively (Fig. 7).

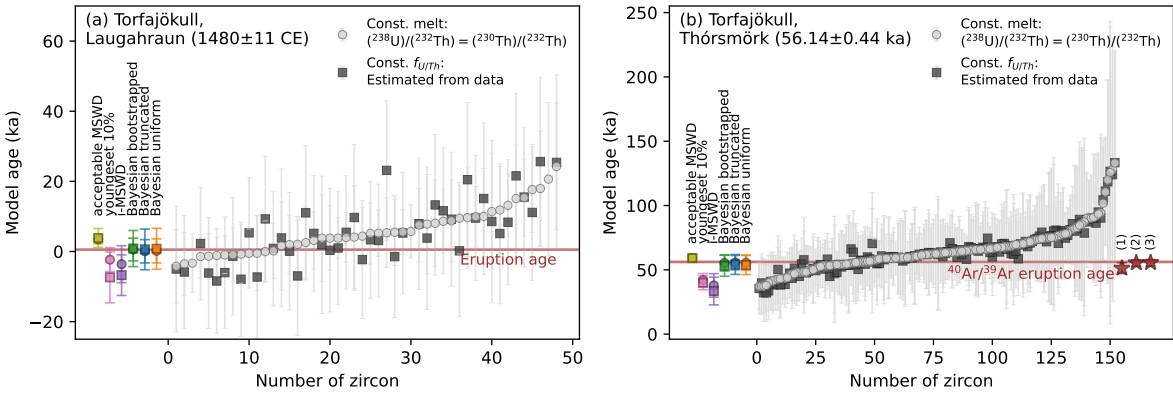

**Figure 7.** Zircon crystallization and eruption age estimates for two validation samples from the Torfajökull volcanic system, Iceland (all uncertainties correspond to $2\sigma$, eruption estimates additionally include a systematic uncertainty of 2.9%). (a-b) Zircon crystallization ages (grey symbols) and eruption age estimates (colored symbols) for two samples from Torfajökull, with the marker shape corresponding to the different model age datasets. The first three estimates represent weighted mean ages calculated from different subsets of zircon ages, whereas the remaining three are Bayesian eruption estimates based on different prior distributions. Measurements of enhanced Th, implying apatite inclusions were ablated along with the zircon, were only used to calculate model ages with the constant melt approach. Shown alongside are suggested eruption ages for Laugahraun based on tephrochronology (Larsen, 1984) and the most recently published $^{40}$Ar/$^{39}$Ar age of glassy fiamme from the Thórsmörk ignimbrite (Groen and Storey, 2022). The stars on the right of (b) correspond to independent eruption age estimates for Thórsmörk: (1) Moles et al. (2019), (2) Svensson et al. (2008), (3) Guillou et al. (2019).

In contrast to Boehnke et al. (2016), we emphasize that using whole rock or groundmass glass as a constant melt anchor point does not generally overestimate ages. However, if the whole rock or groundmass glass is not representative of the melt, as for the Japanese and the Belford Dome sample, then the model ages will be inaccurate. An inappropriate $f_{U/Th}$ can also bias the model ages, either too old for high $f_{U/Th}$ or too young for low $f_{U/Th}$. Therefore, both approaches depend heavily on accurate estimations of either equilibrium melt composition or average $f_{U/Th}$.

### 3.1.5 Recommendations for calculating U-Th model ages

Following the discussion and investigation of the KPT data, as well as the evaluation of the Icelandic and Japanese samples, we can make suggestions on how to retrieve the most reliable zircon crystallization ages from U-Th data. For the constant melt approach, in a first step, the reliability of the measured groundmass glass or whole rock data should be assessed. If a young isochron intercept does not overlap within the uncertainty of the measured glass composition, this suggests that the measured glass is not representative of the melt and should not be used as a melt anchor point. An alternative method would then be to use the young isochron intercept $y_0$ itself as an anchor point. However, if it overlaps the measured glass composition, this gives more confidence that the glass composition represents the equilibrium melt.

In terms of estimating a constant $f_{U/Th}$, we propose a combination of an initial value assessed for the system (based on literature values, direct measurements, chemistry of the system, tuning sample with known eruption age) and approximating it by using the median zircon compositions with the appropriate melt value. This should yield a reasonable estimate for a constant $f_{U/Th}$.

As we have discussed, both approaches represent endmember cases, but for reasonable estimations of the constant equilibrium melt composition or $f_{U/Th}$, the overall model age spectra should be similar and comparable. However, using a constant fractionation factor on a zircon with a strong apatite influence will overestimate its age. Measurements with exceptionally high Th counts or unreasonably low U/Th elemental ratios, evident from Iolite output, should ideally be excluded, as they bias statistical age estimates.

## 3.2 Eruption age estimate using U-Th crystallization ages

### 3.2.1 Accuracy and precision of different eruption age estimation methods

To evaluate the eruption estimate methods, the accuracy and precision for each method were compared. Here we define the accuracy as the difference between the estimated and true (synthetic) eruption age, while precision is defined as the $2\sigma$ uncertainty of the estimated eruption age. We also compare whether the reported uncertainty is realistic relative to the given accuracy, using the percentage of how often the true eruption age falls within the $2\sigma$ uncertainty of the eruption estimate (Fig. 8).

By comparing these parameters across different methods, several interesting observations can be made for the two age ranges we simulated (0-40 ka and 55-170 ka) and as a function of $N_{zircon}$ (10, 30, 50, 70, 90, 110, 130, 150). The accuracy of the eruption age estimates generally increases with $N_{zircon}$. However, for the "acceptable MSWD" weighted mean the accuracy stagnates at a comparable high deviation from the true eruption age, while the "i-MSWD" weighted mean typically overestimates the eruption age for low $N_{zircon}$ while underestimating it for high $N_{zircon}$. The best and most consistent accuracy is achieved by the Bayesian method with the truncated normal or the uniform prior distribution, shortly followed by the bootstrapped prior distribution and the "youngest 10%" weighted mean method. As expected, the accuracy and the precision of all eruption age estimates are better for the young eruption ages than for the older ones, due to lower uncertainties of the underlying data. The precision increases for increasing $N_{zircon}$ for all methods, except for the "i-MSWD" method, which plateaus at a lower $N_{zircon}$. The weighted mean methods tend to underestimate the uncertainty, yielding poorer confidence in the eruption ages. This is most significant for the "acceptable MSWD" weighted mean, as it only captures about 2% of the eruption ages, while the "i-MSWD" weighted mean retrieves the eruption age in 73% of the cases. Best and most consistent performance of the weighted mean methods across the range of $N_{zircon}$ was yielded by the 10% youngest zircon with a confidence of 89%. Generally, Bayesian methods tend to capture the uncertainty more adequately. The prior uniform distribution and the truncated normal distribution perform best and most steadily with high levels of observed confidence of averaging 93% and 92% respectively, while the bootstrapped prior distribution only retrieved 81% of the eruption ages. There is no significant difference in the performance of the truncated normal or uniform prior distribution in the Bayesian method as a function of the underlying

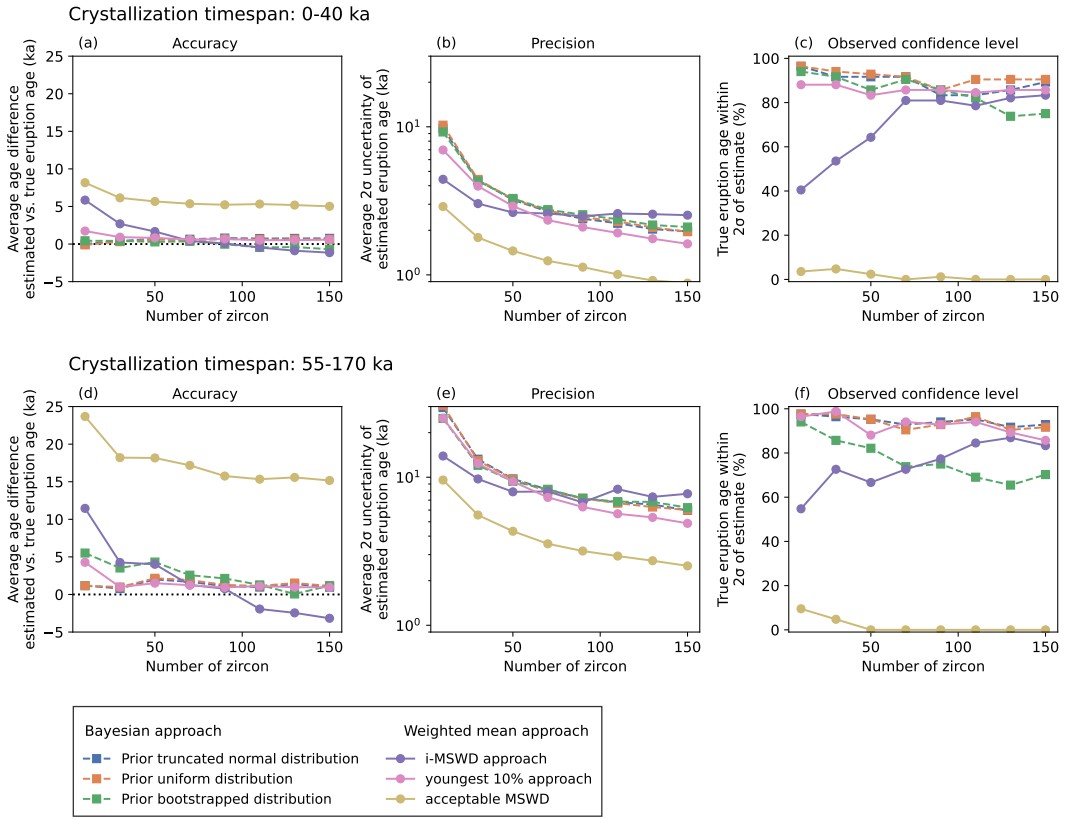

**Figure 8.** Accuracy and precision of different eruption age estimation methods as a function of the number of zircons and two crystallization timescales: (a-c) 0-40 ka and (d-f) 55-170 ka. (a) and (d) Accuracy, expressed as the average difference between the estimated and true eruption ages, with the ideal value of zero indicated by a black dotted line. (b) and (e) Precision, shown as the average $2\sigma$ uncertainty of each method. (c) and (f) Percentage of cases where the true eruption age falls within the $2\sigma$ uncertainty of the estimate.

sampled distribution, which would suggest any bias in favor of the tested prior distributions compared to the weighted mean
method.

### 3.2.2 Optimal method for estimating eruption ages using zircon U-Th data

Our results indicate that Bayesian methods (with uniform and truncated normal distributions as priors) are the best-performing methods for calculating eruption ages using low-dispersed U-Th data, while the weighted mean methods are not satisfactory, similar to the findings of Keller et al. (2018) and Baudry et al. (2024).
The "acceptable MSWD" weighted mean fails to accurately predict the eruption age, which is inherent in the choice of simulating continuous ages within a rather extended timeframe and global MSWDs of mainly 2.5 and higher. It has been shown that those methods overestimate the eruption age significantly for prolonged and continuous crystallization (Keller et al., 2018).

Furthermore, given the increasing number of zircon ages, the uncertainty of the method decreases as it is inversely proportional to the root of the number of samples, which draws a false picture by increasing confidence.

The "i-MSWD" weighted mean method also performed unsatisfactorily. At low $N_{zircon}$, the "i-MSWD" tends to overestimate the eruption age, while for high $N_{zircon}$, it tends to underestimate it. This effect is linked to the relative size of the selected youngest zircon population. With fewer zircons, the proportion of zircon to cause a detectable MSWD jump must be larger, skewing the age estimate older. Conversely, with larger datasets, smaller proportions or ages suffice, often biasing the estimate to younger ages. In the specific case of our synthetic dataset, no distinct young population was implemented, but rather continuous crystallization, similar to what can be resolved in natural datasets for U-Th data. Given the high uncertainties of typical U-Th LA-ICP-MS zircon ages, true young zircon populations are most likely overprinted by Gaussian noise and are unlikely to be resolved. In our synthetic data, any MSWD jumps are randomly generated by Gaussian noise and the given uncertainties and do not reflect underlying geological processes. These jumps are often driven by low uncertainty dates, not by the presence of distinct zircon age populations. Therefore, in the absence of a clear population structure visible in the age plot, visual jumps in the iteratively calculated MSWD are arbitrary and not significant, and influenced by analytical artifacts rather than true geological events. Because the selection is based on visually identified and often ambiguous MSWD changes, this method lacks the objectivity and reproducibility necessary for robust eruption age estimation, particularly in low-dispersed datasets.

The weighted mean method, which seemingly performed best and most consistently for our simulated timescales, was the weighted mean of the 10% youngest zircon ages. The general idea was to use a constant proportion of young zircon ages to respect the statistical need to base the eruption age on multiple data points rather than based on the youngest zircon age alone as well as assuming that the eruption age is best reflected by the young population, while being objective enough to include a constant proportion of the sampled data to avoid systematic shifts. However, here we have only assessed two specific timescales with different levels of uncertainty, which by chance are well described by a weighted mean of the 10% youngest zircon. If the crystallization timescale is shorter and inherently the dispersion gets lower, the best constant percentage will inevitably change towards higher proportions, with an endmember case of instantaneous crystallization allowing for a justified global weighted mean method, while the opposite is true for increasing dispersion (Fig. S4). This illustrates that using a constant proportion of zircon for a weighted mean method is too sensitive to the dispersion of the dataset and therefore will perform inconsistently.

In contrast to the weighted mean methods, which implicitly assume that the selected zircon population represents one single crystallization age (Baudry et al., 2024), the Bayesian method takes into account the entire zircon crystallization dataset with its uncertainties. It still requires an explicit assumption of the underlying age distribution to inform the eruption age calculation. However, as this is in better accordance with true zircon crystallization in a dynamic magmatic system (Schmitt et al., 2023), we prefer the Bayesian over the weighted mean methods.

Previous work (Keller et al., 2018) and our study indicate that the choice of prior can affect the accuracy of the calculated eruption age, but not dramatically for reasonably well-constrained priors that capture the general shape of the underlying zircon crystallization age distribution. Investigation of the most suitable prior for high precision zircon U-Pb datasets suggests a boostrapped prior to be best suitable for well-resolved age dispersion, while a uniform prior is more applicable to low-MSWD datasets (Keller et al., 2018). As expected, for our low-dispersed datasets, the bootstrapped distribution did not perform best.

This is likely partially driven by the increasing uncertainties towards older ages for U-Th datasets, as this inherently distorts the bootstrapping and will not represent an underlying zircon crystallization distribution (Fig. S5). This distortion effect is stronger in the modeled crystallization period of 55-170 ka, explaining why the performance of the bootstrapped prior was poorer for this crystallization period compared to 0-40 ka.

Our analysis shows that both uniform and truncated normal priors perform equally well for U-Th datasets and are useful for retrieving eruption ages. Since most volcanic zircon crystallization is truncated by eruption (Nathwani et al., 2025), a truncated normal prior is reasonable. However, because U-Th uncertainties increase toward older ages, even a dataset drawn from a uniform distribution naturally approximates a truncated normal shape (Fig. S5). With the relatively high uncertainties of U-Th LA-ICP-MS dating, a uniform prior makes fewer assumptions and best reflects periodic zircon growth, where short growth episodes are overprinted by age uncertainties (Fig. S5). We therefore prefer the uniform distribution for volcanic U-Th datasets, in agreement with Baudry et al. (2024).

### 3.2.3 Eruption age estimates of validation samples

The different eruption age estimate methods were also applied to the three natural samples from Japan (Fig. 5) and Iceland with known eruption ages (Fig. 7). The "acceptable MSWD" weighted mean overestimates the eruption age for all three samples. For the Icelandic samples, where the crystallization period is shorter and the ages are less dispersed, the "youngest 10%" weighted mean underestimates the true eruption age, while overestimating it for the strongly dispersed Japanese sample. This highlights that the weighted mean methods are strongly affected by the dispersion of the data (Fig. S4). The "i-MSWD" method strongly underestimates the eruption age of the Thórsmörk sample, but approaches the true eruption age of the other two samples. In accordance with the results from the synthetic datasets, the Bayesian method using the truncated normal and the uniform distribution behaved most consistently. The eruption ages of the Icelandic samples were retrieved, while the eruption age of the Japanese sample is within the uncertainty of the truncated normal Bayesian eruption estimate. A reason for slightly overestimating the eruption age is most likely related to the model age calculation rather than the eruption estimate method. A young isochron intercept slightly higher $1.6 \pm 0.4$, rather than using $1.52 \pm 0.32$ retrieved through the workflow described above, would lower the Bayesian eruption estimates by $\sim$4 ka, more accurately retrieving the true eruption age.

## 3.3 U-Th-Pb double-dating

### 3.3.1 U-Th and U-Pb double-dating of the KPT zircon

By estimating the eruption age of the KPT sample with our preferred eruption age method (truncated normal or uniform Bayesian) for the most agreeable U-Th model age approaches relative to the U-Pb ages (constant melt with secular equilibrium assumption and constant $f_{U/Th}$ of 5), we can test the applicability of our new U-Th-Pb LA-ICP-MS strategy. The U-Th model ages of the KPT, with a crystallization tail between 160-300 ka, have high individual uncertainties, and the MSWD falls well below 1 (Fig. 9). As a result, the Bayesian method fails to recover the published eruption age of 161 ka (Smith et al., 1996), instead converging toward the weighted mean age. These high uncertainties obscure the true crystallization

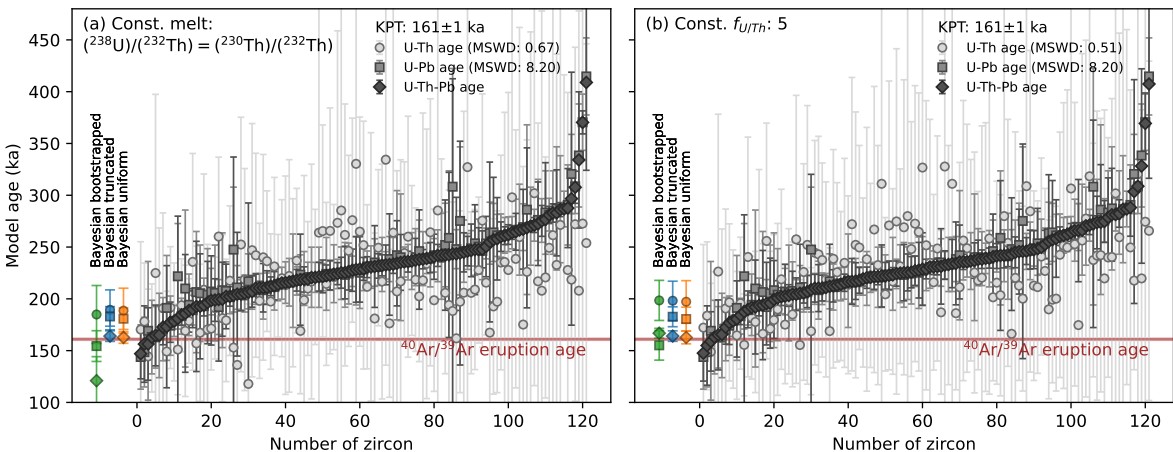

**Figure 9.** Overview of Bayesian eruption age estimates with different prior distributions for the KPT U-Th, U-Pb, and combined U-Th-Pb age datasets, compared to the published eruption age of $161 \pm 1$ ka (Smith et al., 1996). All uncertainties correspond to $2\sigma$. (a) U-Th model ages calculated using the constant melt approach by measuring $(^{238}U)/(^{232}Th)$ and assuming secular equilibrium. (b) U-Th model ages calculated using a constant fractionation factor ($f_{U/Th} = 5$). Only zircon analyses with overlapping U-Th and U-Pb ages within $2\sigma$ were included. Systematic uncertainties of 2.9% 2s (U-Th) and 1.9% 2s (U-Pb) were propagated into the Bayesian eruption age uncertainties. The combined U-Th-Pb ages represent individually weighted mean ages of overlapping U-Th and U-Pb analyses, with weighting including the respective systematic uncertainties. We additionally propagated the average systematic uncertainty of the two systems of 2.4% 2s to the final eruption age estimates using the combined U-Th-Pb ages to avoid underestimating their uncertainties.

distribution and ultimately limit the resolution of the eruption age. Interestingly, the eruption age is also overestimated when using only U-Pb ages, despite the greater age dispersion and a high MSWD of 8.2. This likely reflects the challenges of resolving crystallization ages for the youngest zircons in the U-Pb system, where uncertainties increase due to low radiogenic Pb and the greater influence of common lead corrections. By evaluating both the U-Th and U-Pb ages from the same ablation volume and calculating a weighted mean U-Th-Pb age for each zircon (including their individual systematic uncertainties), greater emphasis is placed on crystals with internally consistent ages. This results in a more robust and well-constrained crystallization age spectrum. Using this combined dataset, the Bayesian method with a uniform or truncated normal prior distribution successfully retrieves the published eruption age of 161 ka within uncertainty for the KPT sample.

### 3.3.2 Applicability of the optimized U-Th-Pb dating routine

The here presented strategy for LA-ICP-MS of simultaneous measurement of U-Th-Pb on zircon works well in resolving U-Th disequilibrium and U-Pb age for the period between roughly 150-300 ka. Within this timeframe, both dating techniques reach their limits. However, with this LA-ICP-MS strategy and the two independent ages, more confidence in their crystallization age can be achieved. On one hand, for younger zircons, the U-Pb age will have increasingly more difficulties resolving the crystallization ages, especially if the zircons are U-poor. However, in this case, the U-Th ages should gain confidence, as they

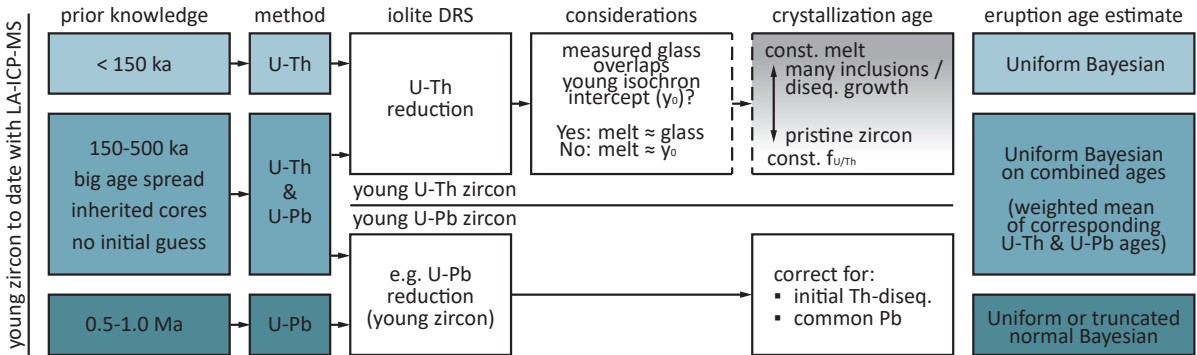

**Figure 10.** Proposed workflow to date young zircon (< 1 Ma) and outline of the applicability of the here presented optimized U-Th-Pb double-dating strategy. The time ranges for different dating strategies are specified, along with the custom iolite DRS that can be applied. The favorable approaches of determining individual zircon crystallization ages are described, including the crucial assessment of whether the measured groundmass glass activity ratio composition overlaps with the young isochron intercept to evaluate the reliability of the groundmass glass measurement. Finally, the recommended method for estimating the eruption age is provided.

diverge further from secular equilibrium. On the other hand, for older zircon crystals already in secular equilibrium, the U-Pb
age gets increasingly better, as more of the radiogenic Pb will reduce the impact of the common lead correction, subsequently inducing more confidence in the U-Pb ages. Therefore, for samples with crystallization ages younger than 150 ka, we suggest using the classic U-Th disequilibrium dating routine (Guillong et al., 2016), as the dwell time on $^{230}$Th is higher and therefore better resolution on the U-Th age can be achieved, especially if the samples are young or U-poor and small amounts of $^{230}$Th have been produced. However, if the samples are older than 150 ka, without knowing the exact age range, or if the samples
frequently contain inherited and recycled cores, this new strategy can add confidence for the younger crystals (via U-Th or combined U-Th and U-Pb ages), while also providing ages for older crystals and their inherited cores via U-Pb (Fig. 10).

### 3.4 Outlook

Significant progress has been made in recent decades toward refining geochronological strategies for young zircon (e.g., Guillong et al., 2014; Sakata et al., 2017), particularly in calculating and interpreting crystallization ages and translating them into
eruption age estimates (Schmitt, 2011; Boehnke et al., 2016; Keller et al., 2018). This study contributes to that ongoing development, while also highlighting several key areas where methodological improvements are still needed. In particular, more statistically robust approaches for handling low-count or zero-count intensities in isotopic measurements could enhance the reliability of crystallization age determinations. For U-Th model ages, developing a generalizable framework that interpolates between the endmember assumptions of constant melt composition and constant fractionation factor, applicable across diverse
sample types, would represent a major advance. A critical component of such a framework would be the accurate and robust estimation of young isochron intercepts based on sample activity ratios to better evaluate the representability of the measured groundmass glass. An alternative way to justify the constant $f_{U/Th}$ endmember approach could be to remove glass and apatite

inclusions from zircon crystals by acid washing prior to analysis. Additionally, the asymmetric nature of uncertainties in U-Th model ages should be better addressed; representing them symmetrically misrepresents the true error structure. While Bayesian methods currently require symmetric uncertainties, using isochron-slopes (rather than ages) as model input provides a promising alternative, as these can be characterized by symmetric errors. However, this shift would also necessitate revisiting prior distributions, as the resulting slope distributions are of a different form than the age distribution and are further dependent on the crystallization timescale (Fig. S6).

## 4 Conclusions

The U-Th-Pb double dating of the KPT zircons and the samples from Iceland and Japan have shown that for reasonable estimates of either the melt composition or the fractionation factor, individual crystallization ages can diverge, but the overall crystallization age spectra look similar. However, in the case of strong influence of apatite inclusions and evidence of disequilibrium growth, the constant melt approach is preferred.

Furthermore, we validated the application of Bayesian eruption age estimates and weighted mean methods for U-Th datasets, which are characterized by increasing uncertainties towards older ages. As the performance of weighted mean methods is highly dependent on the dispersion of the data, we prefer the Bayesian method as it provides superior accuracy and estimates of uncertainties in most cases. Given the typical high individual uncertainties involved in the LA-ICP-MS U-Th dates, we recommend the use of a uniform prior distribution to estimate eruption ages from U-Th zircon datasets.

The combined U-Th-Pb dating strategy of LA-ICP-MS measurements can be applied to any young volcanic zircon below roughly 1 Ma. The U-Th ages can be resolved up to $\sim$300 ka, and U-Pb from $\sim$150 ka onwards, depending on the U content. Consequently, this allows us to gain more confidence in the overlapping period of 150-300 ka, which covers the respective limits of both U-Th and U-Pb dating. Additionally, the ages of inherited cores of young zircon can be retrieved, which would otherwise be hidden by secular equilibrium.

*Code and data availability.* The code and data used in this study are openly available at Zenodo: https://doi.org/10.5281/zenodo.16926790 (Moser, 2025). The repository contains a Python-based Bayesian eruption age estimation function, custom Iolite data reduction schemes, the synthetic U-Th data generator, and a data file containing the LA-ICP-MS data, the calculated model ages, eruption age estimates and technical details on the different LA-ICP-MS strategies.

*Author contributions.* Conceptualisation: ZM, MG, CN, OB. Data curation: ZM, KI. Formal analysis: ZM. Software: ZM, MG, CN. Investigation: ZM, MG. Methodology: ZM, MG. Resources: OB. Writing (original draft preparation): ZM. Writing (review and editing): MG, CN, KI, RGP, OB.

*Competing interests.* The authors declare no conflicts of interest related to this work. We have no commercial or financial relationships that could be construed as a potential conflict of interest.

*Acknowledgements.* This work was supported by the Swiss National Science Foundation (grant 214930). Thanks to Iolite for providing a free student licence. We thank Sæmundur Ari Halldórsson and Kristján Jónasson for their contributions during the fieldwork in Iceland. We thank Dawid Szymanowski and Francesca Forni for helpful discussions. We thank the two anonymous reviewers and Ryan Ickert, the editor, for their constructive feedback, which helped improve the manuscript.

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
