# Peer review of "Improving crystallization and eruption age estimation using U-Th disequilibrium dating of young volcanic zircon"

_EGUsphere, 2025_

## Author Response (AR1)

**Authors' reply to Editor of**

**Improving crystallization and eruption age estimation using U–Th and U-Pb dating of young volcanic zircon**

Zoe Moser, Marcel Guillong, Chetan Nathwani, Kurumi Iwahashi, Razvan-Gabriel Popa, and Olivier Bachmann
*Geochronology Discussion:* https://doi.org/10.5194/egusphere-2025-4232
* * *
**EC**: Editors' Comment, **AR**: Authors' Response

Public justification (visible to the public if the article is accepted and published):

**EC:** Thank you to the authors for their submission and to Anonymous Reviewer 1 and Anonymous Reviewer 2 for their reviews. In particular, as Reviewer 2 pointed out, the constructive feedback of Reviewer 1 was particularly thorough. Thank you to the authors for their constructive responses to the Reviewers comments. This manuscript is a useful addition to the growing body of literature on how to address the complicated issue of variability in crystallization ages for mineral chronometer dates and may be suitable for publication in Geochronology following revisions.

Additional private note (visible to authors and reviewers only):

**EC:** Thanks to everyone (reviewers and authors) for the constructive discussion.

**EC:** To the authors: Thanks for engaging thoughtfully with the detailed review from Reviewer 1. As you revise the manuscript, please consider that even in places where you may disagree with the comments - and if your disagreement is correct - the reviewer, who is a peer and an expert, put a lot of time and effort into reading and trying to understand your mansucript, and if they come to the wrong conclusion, it may be becasue your manuscript requires clarification. I would urge you to re-read the comments in that light and take the opportunity to clarify and strenghten your arguments in places where you disagree with the reviewer.

**AR:** Thanks for pointing out the need for clarification. We went back to the reviewer comments and added more details where we were not agreeing with Reviewer 1 to avoid any misunderstandings.

Regarding Reviewer 1 comment on the melt anchor point: we added a more detailed definition.

"...one idea assumes that a constant isotopic melt composition in terms of $(^{230}\text{Th})/(^{232}\text{Th})$ and $(^{238}\text{U})/(^{232}\text{Th})$, hereafter referred to as the melt anchor point, can be linked to the individual zircon isotopic compositions (Schmitt, 2011),..."

Regarding Reviewer 1 comment on SIMS versus LA-ICP-MS: We have now included a brief statement in the manuscript explaining why SIMS and LA-ICP-MS data can be used interchangeably in our modeling framework.

"... Since these published data are used solely to validate the model age approaches, activity ratios obtained by either SIMS or LA-ICP-MS can be treated equivalently, because the analytical differences do not influence their use in the subsequent data treatment."

Regarding Reviewer 1 comment on line 56: We added a small clarification in the manuscript to the growing uncertainty of the U-Th ages for older ages.

"... In U-Th disequilibrium dating, uncertainties increase for older ages due to the exponential convergence toward secular equilibrium, making the zircon-melt isochron age increasingly sensitive to analytical uncertainties in the activity ratios (Schmitt, 2011). ...

**EC:** There are no objections to the scientific content, but I suspect that a revised manuscript might need editorial streamlining, so I am recommending that the revised ms might be reviewed by the edtior (myself). By keeping delays in external reviews out of the way, this will permit the next part of the editorial process ot conclude much faster. Please do not be concerned that I am trying to keep it from publication, I just want it to be as good a manuscript as possible.

**AR:** We thank the editor for this suggestion and fully support the idea of editorial streamlining. We appreciate the efforts to help improve the clarity and presentation of the manuscript, and we welcome any guidance in the revised version.

**EC:** One place that I recommend that you pay particular attention to is to carefully define (and if necessary, re-define) parameters in the text. There are references to U/Th, F_U/Th, activity ratios, isotope ratios, mass ratios etc. and it does become somewhat confusing and difficult to read. F_U/Th in particular doesn't appear to be clearly defined, despite its importance. Part of the confusion there is that the quantity can be derived from ether partition coefficient ratios or ratios of elemental ratios - it's probably worth spending some time explaining that in the text for the benefit of readers. Please also define that you are using parentheses to represent activity ratios. Of course, this is a common issue in many manuscripts and is the result of authors reading the same few paragraphs over and over and having a difficult time seeing them with fresh eyes - it's not a comment on the quality of the manuscript or the skill of the authors. Please consider re-define something in the text when it occurs in a figure caption (which readers will often examine without reading the full text) or use opportunities in the text to revisit a definition that occurred much earlier, particularly if it occurs with other parameters.

**AR:** Thanks for stating the need for clarification. We added some important definitions:

"On the other hand, a constant U/Th fractionation factor ($f_{U/Th}$) between zircon and melt can be assumed (Boehnke et al., 2016), where $f_{U/Th}$ is defined as the ratio of the uranium-to-thorium partition coefficients D between zircon and melt:

$$f_{U/Th} = \frac{D^U_{zircon-melt}}{D^{Th}_{zircon-melt}} = \frac{(U/Th)_{zircon}}{(U/Th)_{melt}},$$ (1)

which corresponds to the ratio of the U/Th elemental ratios in zircon and the melt in equilibrium with the zircon (similar to Sakata et al., 2017)."

"To calculate these ages, each zircon analysis, reported as ratios of isotopic activities expressed using parantheses $(^{230}Th)/(^{232}Th)$ and $(^{238}U)/(^{232}Th)$ (Reid et al., 1997), needs to be linked to the melt from which it crystallized to form a two-point zircon-melt isochron."

We also made sure to state it clearly, where U/Th refers to elemental ratios.

**EC:** Please use μg/g rather than the disclipline-specific convention of "ppm".

**AR:** Thanks for mentioning, we changed this accordingly in the manuscript.

**EC:** Please ensure there are spaces between quantities and units and between quantities and dyadic operators (+, -, ±). There are a few cases where there are no spaces, particularly around ± signs.

**AR:** Thanks for this hint. We changed it throughout the manuscript. We also ensured spaces between quantities and > or <.

Additional comments that are in specific response to your responses to Reviewer 1:

**EC:** Line 41. I don't understand your response to the reviewer. The manuscript states: "To directly date the eruption, alternative chronometers such as zircon (U-Th)/He thermochronology can be employed". It seems obvious to me that Ar/Ar could also be employed, as it is a robust, well understood chronometer, regularly used to date eruptive events. U-Th/He is suitable in some specific instances but it's certainly not as precise as Ar/Ar and not as widely applicable. There are many other techniques available - He-3 cosmogenic burial dating, C-14 dating of buried organics, but it's peculiar that you only wish to mention zircon-based techniques. Perhaps the intention of the paragraph is something different than the way it is being read by me and the reviewer, but either be more inclusive or more clear.

**AR:** We did not mean to be dismissive of the suggestion, we simply wanted to focus on zircon-specific methods to not divert too much from the main point of the paper. But we understand your point, and we included some additional methods to be more inclusive.

"... To directly date the eruption, alternative chronometers such as zircon (U-Th)/He thermochronology, $^{40}Ar/^{39}Ar$ dating or radiocarbon dating can be employed. (1) Zircon (U-Th)/He thermochronology

records the time of cooling below the closure temperature of He of ∼200°C (Reiners et al., 2002), which often corresponds closely to the eruption age (e.g., Friedrichs et al., 2020). This dating technique comes however with many challenges, including the need for corrections related to initial $^{230}$Th-$^{238}$U disequilibrium, alpha-ejection effects, pre-eruptive residence time, and accounting for internal age and compositional heterogeneity (Friedrichs et al., 2020). (2) $^{40}$Ar/$^{39}$Ar dating is widely used on potassium-rich minerals or volcanic glass (e.g., Smith et al., 1996; Groen and Storey, 2022; Castellanos Melendez et al., 2023). This technique typically records the time when argon becomes trapped within the mineral or glass, which is commonly assumed to coincide with eruption (Kelley, 2002a). However, the presence of excess or inherited argon, as well as post-eruption alteration, can bias the resulting ages, complicating their interpretation (Kelley, 2002b; Ellis et al., 2017). (3) Radiocarbon dating of organic material buried or killed by volcanic deposits provides another means of constraining the timing of eruption (e.g., Friedrich et al., 2006; Danišík et al., 2020). Its applicability is limited to ages younger than ∼55 ka (Hajdas et al., 2021), and it requires the presence of suitable organic horizons directly beneath volcanic deposits, conditions that may be difficult to meet when attempting extensive and comparable dating across a volcanic system. Alternatively, we can infer the eruption age from the latest zircon crystallization, if we assume that zircon crystallized until the eruption took place (Keller et al., 2018; Nathwani et al., 2025)....."

**EC:** Line 112, comment regarding f_Th/U "fractionation factor". The reviewers comment points out that it's confusing. It looks to me like this appears in the abstract and is perhaps not defined, at least not before the line 112 paragraph. Your nomenclature is ok but you need to explicitly define that you are referring specifically to ratios of partition coefficients.

**AR:** Thanks for this comment. We defined the variable according to the comment above and referred to the equation.

**EC:** Line 145: Your response is confusing. If the method is in development, it is not ready to reduce analytical data. You have clearly just lightly edited text from the Vermeesch and Gloire abstract rather than written something meant to explain the data treatment approach to the readers of this paper. The Vermeesch abstract is incomprehensible to the typical reader of your manuscript, so please rephrase in such a way that makes sense. Describe the real (not a potential or imaginary, as described in the vermeesch abstract) problem associated with a specific data treatment procedure that is commonly used, where the problem affects scientific inference and not just a perception of inaccuracy untethered from actual measurements and the study of real natural samples. Then describe the quantitative difference between that, problematic method, and the new method.

**AR:** We appreciate this comment. Our intention in the original text was to briefly highlight emerging developments and potential future improvements in data-reduction methodology. However, we agree that including these ideas here may confuse readers, especially as the approach is still in development. To maintain clarity and to avoid referencing methods that are not yet published or fully validated, we have removed this section from the revised manuscript.

**EC:** Line 153: In your comments, ensure there is a space between the quantity and the unit (55 ka).

**AR:** Thanks for pointing that out, we changed it accordingly.

**EC:** Line 238: Please eliminate the sentence starting "The oxygen fugacity of the system further...". See, for example Liang et al. (2025; GCA https://doi.org/10.1016/j.gca.2025.06.027) but essentially, the Burnham and Berry experiments that had large shifts in DU/DTh at high fO2 are not relevant to natural systems. Those partition coefficient ratios would result in, for example, 230-Th *excesses* in zircon, which to my knowledge are never seen. Whether or not the Burnham and Berry experiments at high- fO2 are accurate, there is no evidence that oxygen fugacity is a major factor in U-Th fractionation in zircon.

**AR:** Thanks for highlighting this new contribution. We deleted this sentence accordingly, and added a short comment acknowledging the new contribution.

"... Beyond crystal-scale effects, it has also been shown that $f_{U/Th}$ tends to decrease with increasing crystallization temperatures (Liang et al., 2025), while $f_{U/Th}$ was shown to decrease with increasing silica in the whole rock (Kirkland et al., 2015). ..."

**EC:** Line 286: It seems to me that while you are correct that there is no reason to assume that zircon crystallization stopped in the crystal-rich portions of the melt, you've given no reason to assume that it will keep going. Please either expand your argument or follow the reviewers recommendation to consider this option in the text (my preference is the latter, given the lack of evidence presented, but perhaps there is information I do not have that suggests that crystallization continues until eruption).

**AR:** We agree that, given the available evidence, it cannot be conclusively excluded that zircon crystallization stopped before the eruption in the sampled portions of the crystal mush. We have therefore revised the text to reflect this uncertainty and to acknowledge more explicitly the possibility raised by the reviewer.

"There is no evidence for zircon resorption that could explain the gap between the eruption age and the latest recorded zircon crystallization, nor is there any indication that zircon growth had ceased. Although it cannot be determined conclusively whether zircon continued to crystallize until the time of eruption, the evolution plot shows that the low-$(^{230}\text{Th})/(^{232}\text{Th})$ data form an almost horizontal array, consistent with a young, near-zero-age isochron for this historic sample."

"Such mixing likely remobilized zircon from the crystal mush, consistent with the observed long zircon crystallization timescales, and caused significant changes in melt composition, so that the erupted melt was no longer in equilibrium with any of the zircons from this sample."

**EC:** Figure 5: It's not obvious to me why you are not also using the Barboni et al data. It's not acceptable to ignore that data because it was not used in the Boehnke et al arguments, rather, the full dataset should be used to refine or adjust those arguments as appropriate. I very strongly advise you to include the dataset in your manuscript. If you decide not to, you must explain *in the text of the manuscript* very clearly why the Barboni data is not included.

**AR:** We now include the data by Barboni et al. (2016). It did not change or alter any conclusions, except that we had to adjust the young isochron intercept slightly from 1.5 to 1.3. We adjusted Figure 6 and the supplementary Excel file accordingly.

**References**

[revised manuscript text omitted]

Zoe Moser, Marcel Guillong, Chetan Nathwani, Kurumi Iwahashi, Razvan-Gabriel Popa, and Olivier Bachmann
*Geochronology Discussion:* https://doi.org/10.5194/egusphere-2025-4232-RC2
* * *
**RC**: Reviewers' Comment, **AR**: Authors' Response

**RC:** The problem Moser and coauthors address is a difficult and rather statistically complicated one, of a type that is becoming increasingly common in geochronology – that is, how to interpret dispersed mineral crystallization age datasets with complicated uncertainty structures, in this case from two different systems (U/Th and (U-Th)/Pb) at once. While there is probably (as usual) more that could be done, I think this manuscript presents a generally reasonable approach to the problem, and should make a good contribution to the literature.

**AR:** We thank the anonymous referee #2 for their constructive suggestions. We appreciate the input and have improved the manuscript according to our comments below.

**RC:** I agree with just about all of the comments made in the very thorough review by Reviewer 1, so will not belabor the point by repeating them. If the authors wish to continue to emphasize the analytical methods aspect of joint U/Th and (U-Th)/Pb measurement, I don't object in principle but would ask for more information in the main text about the technical details of this joint measurement, and how it compares to previous attempts at such the Ito papers mentioned by Reviewer 1.

**AR:** Thanks for this comment. We understand and agree with the necessity of comparing our approach with Ito's approach.

"Our applied U-Th-Pb LA-ICP-MS routine follows the general idea of Ito (2014, 2024), but differs in the optimization of dwell times to improve precision on minor isotopes (e.g., $^{206}Pb$, $^{207}Pb$, $^{230}Th$) and in the selection of measured masses to allow direct mass bias correction by including $^{235}U$ while avoiding measurements of masses not required for our correction scheme (202, 204, 208)."

"In contrast to the approach by Ito (2014, 2024), who modified the U–Pb dating protocol to include Th measurements, we adapted the U–Th dating protocol of Guillong et al. (2016) to additionally measure Pb."

We added more details for strengthening our optimized U-Th-Pb LA-ICP-MS strategy, and explained the reasoning for the chosen parameters. In terms of the technical details, also regarding the comments of Reviewer 1, we added the following:

(1) We clarified how the AFC analog counting factor was determined.

(2) We explained the reasoning for alternating the magnet mass position: "For the U–Th–Pb LA-ICP-MS strategy, the magnet mass position was alternated between the low-mass Pb peaks (206, 207) and the higher-mass U and Th peaks ($\geq$230), to minimize non-linear mass bias introduced by magnetic dispersion across the mass range."

(3) We elaborated on our sample-standard bracketing protocol.

(4) We specified our validation approach.

(5) We added information about our data processing, corrections applied, and uncertainty propagation.

Additionally, we added a sheet in the supplementary Excel file containing more detailed operating parameters for the differently used LA-ICP-MS strategies.

**RC:** Beyond what has already been covered by Reviewer 1, my main request would be to include more discussion of the magnitude and treatment of systematic uncertainties (which generally should be excluded during Bayesian age estimation, and then re-applied to the result), particularly in the context

of combining U/Th and (U-Th)/Pb measurements which have quite disparate systematic uncertainty structures. Doing this perfectly is a hard problem which I don't expect the authors to solve completely, but I think some additional consideration is warranted.

**AR:** Thank you for highlighting the importance of discussing systematic uncertainties. We have now included a dedicated section on this topic in the Methods. In general, the analytical uncertainties of our measurements are relatively large compared to the systematic uncertainties, which means the systematic uncertainties only represent a minor contribution to the total uncertainty. Nevertheless, we agree that a discussion of their treatment is valuable.

Unfortunately, there is not yet a quantified assessment of the long term variance of validation reference material for U–Th dating in our laboratory, or in the literature. We agree that such an assessment would be valuable and could be achieved through continuous measurement of one or more well-characterized reference materials. Ideally, these reference materials should be available in sufficient amounts, exhibit near-ideal isochron behavior, and cover a suitable range of U/Th ratios to allow precise determination of the reproducibility of the isochron age within our and possibly other laboratories. This can not be achieved in the context of this study, which is why we had to make assumptions about the long-term variance.

As pointed out by Reviewer 2, the treatment of systematic uncertainties is particularly challenging when combining U–Th and U–Pb ages, and in subsequent eruption age estimations. We added a discussion about this in the methods as well. We are aware that it is a simplification, but it seems to be the most reasonable approach.

[revised manuscript text omitted]

*Geochronology Discussion:* https://doi.org/10.5194/egusphere-2025-4232-RC1
* * *
**RC**: Reviewers' Comment, **AR**: Authors' Response

General Comments for Moser et al manuscript:

**RC:** This manuscript critically examines the assumptions underlying the calculation of 230Th/238U zircon isochron model ages and explores how these crystallization ages can be used to estimate eruption ages. The authors specifically test previously established models for estimating eruption ages from U-Pb zircon datasets, applying them to U-Th datasets. While the manuscript also suggests the introduction of a new U-Th-Pb methodology, I recommend de-emphasizing this aspect, as it detracts from the primary focus and main contribution of the work, which is age modeling. Additionally, this is not the first study to propose LA-ICPMS U-Th-Pb dating (see Ito publications). If the authors intend to present this as a methods paper, more substantial documentation of the method development would be needed. Throughout the manuscript, I found myself seeking more detailed explanations of the methods and model parameters, which I have outlined below. The manuscript would also benefit from editorial improvements, including clearer terminology, improved sentence structure, additional context for arguments, and more detailed figure captions.

**AR:** We thank the anonymous referee #1 for their constructive comments on our manuscript. We appreciate the input and have improved the manuscript according to our comments below.

We have expanded the text to clarify how our U–Th–Pb double-dating strategy differs from previously published approaches (Ito, 2014, 2024). In addition, we have provided further methodological details concerning measurement procedures, data corrections, and error propagation. However, as the primary focus of this study lies in the determination of U–Th model ages from measured activity ratios, the optimized U–Th–Pb LA-ICP-MS strategy is presented mainly in the context of supporting and discussing this main aspect.

Specific Comments:

**RC:** As currently written, the manuscript devotes the vast majority of its discussion (approximately 95%) to 230Th/238U, with only minimal attention given to 206Pb/238U, which appears as an afterthought. Therefore, I find it an overstatement to claim that the paper presents a new U-Th-Pb dating technique, as the primary focus is clearly on age modeling. Positioning this as a methods paper diminishes its more significant contribution: refining the calculation of isochron 230Th/238U model ages. I recommend removing U-Pb from the title, de-emphasizing the methods aspect in the introduction, and revising the third section of the discussion accordingly. Furthermore, the claim of being the first to propose simultaneous U-Th-Pb dating is inaccurate, as similar approaches have been published previously (see Ito, 2014; Ito, 2024). It could be worthwhile commenting how your methods compare to Ito. I also noticed some circular reasoning in the use of U-Th and U-Pb ages, where the selection of an appropriate DTh/U for the initial 230Th disequilibrium correction appears to be based on achieving consistency between the resulting U-Pb and U-Th ages. Similarly, the choice of melt composition model is determined by whether the U-Th ages match the U-Pb ages.

**AR:** Thank you for pointing out these specific papers. We want to stress that we never claimed to be the first ones to do double-dating on zircon. We intended to present a new LA-ICP-MS dating strategy, particularly optimizing the measured dwell times and needed measured masses, to be able to accurately do the data reduction for both dating techniques, including all necessary corrections. But indeed, we agree that it is important to comment on how our method differs from Ito's (2014; 2024) precariously published suggestions.

"... Our applied U-Th-Pb LA-ICP-MS routine follows the general idea of Ito (2014, 2024), but differs in the optimization of dwell times to improve precision on minor isotopes (e.g., $^{206}$Pb, $^{207}$Pb, $^{230}$Th) and in the selection of measured masses to allow direct mass bias correction by including $^{235}$U while avoiding measurements of masses not required for our correction scheme (202, 204, 208). ..."

We are considering changing the title of the manuscript to avoid the term U-Pb, and we adjusted the abstract to make it clear that we are not proposing a new dating technique but that we apply an optimized LA-ICP-MS strategy.

"...we applied an optimized LA-ICP-MS U–Th–Pb double-dating strategy that simultaneously retrieves U–Th and U–Pb ages from the same zircon ablation volume. ..."

We added in the text the reasoning for the used $f_{Th/U} = 0.2$.

"... Initial disequilibrium was corrected by assuming a Th/U fractionation factor of $0.20 \pm 0.04$ between zircon and melt, which lies between the published values of 0.25 and 0.18 for the KPT (Guillong et al., 2014). Using a Th/U fractionation factor of 0.25 to correct for initial Th disequilibrium would yield ages roughly 5 ky younger. ..."

But indeed, we are aware of the circularity, particularly if the U-Th ages are calculated by applying a constant $f_{U/Th}$. We discussed this in section 3.1.3.

**RC:** Given the presence of multiple U-Th and U-Pb chronometers, it would be helpful to clearly specify which chronometers (230Th/238U and 206Pb/238U) are the main focus of this paper. You could either use the full isotope notation consistently throughout or define each chronometer at the outset and use abbreviations thereafter. This distinction is important, as you also use U/Th to refer to elemental ratios, and in the methods section, "U-Th" refers to multiple isotopes of each element. The manuscript would benefit from a stronger introduction outlining the necessary corrections for U-Th and U-Pb dating. One major challenge in U-Pb dating of zircon in this age range ($< 1$ Ma) is the correction for initial 230Th disequilibrium. The text does not adequately explain why this correction is needed, how it is performed, and why it is particularly important for young zircon ($< 1$ Ma). In your discussion (pg. 11), you note that variability in the initial 230Th/232Th ratio, extrapolated from glass measurements, can significantly affect zircon crystallization ages. It would be useful to clarify whether variability in the initial 230Th/232Th ratio or analytical precision has a greater impact on age variability, and whether this effect changes depending on the age of the zircon.

**AR:** Thanks for noting that. We defined in the introduction, which isotopic system we are referring to. We highlighted the necessary corrections in the methodology part for each dating technique. And we added some more detail regarding the disequilibrium correction for U-Pb dating.

"... These crystals start with a U–Th disequilibrium at the time of crystallization due to U/Th fractionation. This disequilibrium is utilized when applying U–Th disequilibrium dating. However, to quantify the U–Pb age, it is necessary to correct for the initial $^{230}$Th deficit to obtain accurate $^{238}$U–$^{206}$Pb ages (e.g., Schärer, 1984; Sakata et al., 2017). ..."

The variability of the measured 230Th/232Th in groundmass glass is mainly owed to the lower analytical precision (higher uncertainties) and not to true variability in the glass (at least when it is fresh and homogeneous). The effect of inaccurate groundmass glass composition is illustrated in Fig. S2 for three zircons with different ages.

**RC:** There are a couple of instances where references are light or predominantly cite co-authors or researchers affiliated with ETH Zurich, while overlooking significant contributions from other individuals and laboratories. Including a broader range of references would more accurately reflect the breadth of work in this field and acknowledge the efforts of the wider scientific community. I made note in a few places, but I would check all of your citations. It is generally preferable to include more citations rather than risk omitting important contributions, as this helps foster inclusivity and ensures proper recognition of prior research.

**AR:** We fully agree with and appreciate this comment. We understand the importance of acknowledging the broader scientific community and ensuring that relevant contributions from other researchers

and laboratories are properly cited. In terms of methodology, it is somewhat challenging, as the work relies on specific procedures and analytical developments that have been refined within our laboratory over many years. Nevertheless, we will carefully review all references and add citations where appropriate to ensure the manuscript reflects the full scope of relevant work in the field.

**RC:** Throughout the manuscript, the authors use several vague terms that could be clarified to improve the overall readability. For instance, phrases such as "zircon signal," "crystallization bloom," "zircon subsets," and "melt anchor point" are ambiguous. I am unsure of the intended meaning for most of them and recommend either replacing these terms with more precise language or providing explicit definitions upon first use. The term "melt anchor point" should be rephrased to "initial 230Th/232Th ratio" for clarity. Additionally, the manuscript frequently uses "techniques," "methods," and "approaches" interchangeably, which sometimes makes it unclear whether you are referring to age modeling or analytical procedures. This issue is compounded when the text shifts between these concepts within the same paragraph (for example, in line 60). I suggest reviewing the manuscript for terminology consistency throughout.

**AR:** Thank you for pointing out the need for clarity in terminology. We have revised the manuscript to either replace ambiguous terms for clarity (e.g., zircon signal, crystallization bloom, zircon subsets) or provide explicit definitions upon first use (e.g., melt anchor point). As the melt anchor point refers to the constant isotopic melt composition, which includes not only the initial 230Th/232Th ratio but also 238U/232Th, we consider it inappropriate to rephrase it solely as the initial 230Th/232Th ratio.

For consistency, we have standardized terminology throughout the manuscript as follows:

Techniques: U–Th dating, U–Pb dating

New LA-ICP-MS strategy: U–Th–Pb double dating measurement

Typical LA-ICP-MS routine: U–Th routine, U–Pb routine

Methods: consistently used for eruption age determination methods

Approaches: consistently used for model age calculation approaches

**RC:** In the Samples section, please clearly specify which samples were analyzed using LA-ICPMS and which published datasets were selected for age modeling. Currently, this information is only apparent after reviewing the raw data tables, which makes it difficult for readers to follow. Furthermore, it appears that some of the published data used for age modeling were generated by SIMS, while others were obtained via LA-ICPMS. Additionally, a brief explanation of the rationale for including both LA-ICPMS and SIMS data to test your models would be helpful. For reference, Barboni et al. (2016) also includes SIMS 230Th/238U zircon dates for the Belford dome, which may be relevant to your discussion.

**AR:** Thank you for pointing out that this was not sufficiently clear. We have now specified more explicitly in the Samples section which samples were analyzed using which LA-ICP-MS strategy.

"... for samples with well-constrained eruption ages. For this purpose, we used the classical U-Th LA-ICP-MS measurement routine (Guillong et al., 2016). ..."

While SIMS and LA-ICP-MS differ in analytical setup, there is no fundamental difference in how the resulting isotopic data are used for age modeling in this study. Therefore, we consider it unnecessary to discuss these methodological differences in detail here.

The datasets included in the modeling were chosen to match those analyzed by Boehnke et al. (2016) to facilitate a direct comparison with their results. For this reason, additional datasets (e.g. Barboni et al., 2016) were not incorporated.

**RC:** There a several studies that suggest the importance of applying chemical abrasion treatment to zircon before LA-ICPMS U-Pb dating (e.g., Crowley et al., 2014; Von Quadt et al., 2014; McKanna et al., 2023; Donaghy et al., 2024). Although Pb loss is very unlikely in < 1 Ma zircon, the zircon standards in this study are quite old and will have experienced significant radiation damage. How did the authors handle this issue?

**AR:** We used well-known zircon standards that are referenced in the manuscript, which generally do not show significant Pb loss and are typically concordant. Since our young measured unknown samples were not treated with chemical abrasion or annealing, we also used non-treated reference materials. As we are working with very young zircon, where the biggest uncertainty factors are: the counting statistics uncertainty on 206Pb, the common Pb correction and the Th disequilibrium correction. The uncertainty/bias that might be induced by not applying chemical abrasion and annealing to the standard zircon is negligible.

**RC:** The Methods section would benefit from additional detail and clarification. Since in-situ techniques depend on sample-standard bracketing, it would be helpful to describe how standards were run, including the frequency of standard measurements (e.g., were standards measured only at the start of each run, or throughout?). Please provide information on how corrections were made for abundance sensitivity, relative sensitivity, and drift, or cite previous methods if these were followed. If 91500 was used to quantify concentrations, please specify the assumed concentrations for this standard. Additionally, clarification on how low 230Th counts were handled would be useful. I noticed that the 230Th background increases progressively for both unknowns and standards during each LA-ICPMS session; any insight into the cause of this trend would be appreciated. In some cases (for example, lines 86–98 in the table for 91500), the 230Th background is higher than for the standard, yet the corrected 230Th value remains positive—could you explain how the corrected 230Th is calculated in these instances? The (230Th)/(238U) ratios in your secular equilibrium standards show considerable variation, although the uncertainties (7–12%, 2 sigma) mean they still overlap with secular equilibrium. A brief discussion of this variability and its implications would strengthen the Methods section. Please also provide a similar discussion or relevant citation for your U-Pb analyses. Additionally, if authors want to emphasize new approaches to the U-Th-Pb LA-ICPMS technique, there needs to be more details about how uncertainties are propagated.

**AR:** We added the sample-standard bracketing protocol for the zircon and groundmass glass measurements.

"... All zircon measurements were conducted using a sample–standard bracketing protocol. Each analytical session began and ended with two consecutive analyses of the primary standards NIST612 and zircon 91500. These standards were repeatedly measured in duplicate at regular intervals of 20–30 unknown analyses throughout each session. The zircon blank and secondary reference zircons were analyzed as single analyses at similar intervals but offset from the primary standards. The monazite was measured at the beginning, midpoint, and end of each session. At each time point, it was analyzed using three different ablation parameter sets (9 µm, 2 Hz; 9 µm, 3 Hz; and 13 µm, 3 Hz) to assess the abundance sensitivity of $^{232}$Th on $^{230}$Th with respect to different intensities, and to assess possible drift during a session. The protocol for the groundmass glass was similar, except that only NIST612 was measured in duplicate, while the other validation reference glasses were analyzed as single measurement points. ..."

We put the reference in the text again of the U-Th data processing routine.

We added the concentrations of the 91500 zircon of U and Th.

"... Zircon Th and U concentrations were estimated relative to the reference zircon 91500, with assumed values of 80 ppm U and 30 ppm Th (Wiedenbeck et al., 1995). ..."

We assume that the progressive increase in the $^{230}$Th background results from the precipitation and accumulation of zircon material onto the skimmer cone during prolonged zircon ablation, followed by partial remobilization by the extraction lens at high voltage during the session. We observed that cleaning the skimmer cone before each analytical session significantly reduces the initial background. However, as this remains an empirical observation rather than a systematically tested effect, we have not included it in the manuscript.

The CPS in the datatables are all background corrected except for the background itself.

Similar to the U-Pb data processing for low-count statistics, we calculate the 230Th/232Th and 230Th/238U ratios through integrated intensities (ROI) rather than the mean of individual point-by-point ratios (MOR). As mentioned in the supplementary PDF, the detection limit is calculated after Tanner (2010). The measurements for which the corrected 230Th fall below the limit of detection are not further considered.

We added a short comment in the text in regards to the secular equilibrium of the reference zircon, and we added a figure to the supplementary file.

"... Throughout the sessions, a mean value of 1.002 (median of 0.994) was achieved, with a trend of higher variability and uncertainties for U-poor reference zircons (Fig. S1). ..."

We also added a short comment on uncertainty propagation in our U-Pb DRS.

"... Uncertainties were propagated using analytical Gaussian error propagation, combining the uncertainties of the isotope ratios and correction factors. ..."

**RC:** It was unclear from the Methods section which samples were analyzed in this study and which were sourced from the literature. Explicitly stating which samples were analyzed by LA-ICPMS and clarifying which were measured using the U-Th-Pb method versus the U-Th method would improve clarity. Additionally, please describe how errors were propagated (e.g., quadrature, linearized uncertainty propagation, or Monte Carlo methods), and provide details on sample preparation for LA-ICPMS analysis (such as sticky mount or epoxy mount). Finally, I only discovered at the end of the manuscript that the raw data are available in a separate repository. It would be helpful to mention this in the Methods section. Including the raw data spreadsheet in the supplementary materials would also be beneficial, though I am not certain of the GChron guidelines regarding supplementary files if the data has been posted in a separate repository.

**AR:** Thanks for pointing out that it was not clear. We added to Table 1, which samples were analysed with which LA-ICP-MS routine, and we specified this in the text.

"... We then validate these model age approaches by applying the best model age calculation methods to U–Th datasets for samples with well-constrained eruption ages. For this purpose, we used the classical U–Th LA-ICP-MS measurement routine (Guillong et al., 2016). ..."

We added details to sample preparation.

"... The samples were prepared by crushing using a high-voltage selective fragmentation (SELFRAG) apparatus. Zircon crystals were then concentrated by heavy liquid separation with a sodium-polytungstate solution. Zircon crystals and groundmass glass shards were handpicked and mounted in epoxy resin. The grains were subsequently exposed by grinding and polished with a diamond suspension. ..."

We added a comment to mention that the data is openly available.

"... The data is available in the supplementary files. ..."

We also added some more detailed information as to how we propagated the uncertainties.

"... The uncertainties on the measured $^{238}$U/$^{232}$Th and $^{230}$Th/$^{232}$Th ratios for zircon and groundmass glass, as well as the uncertainty on $fU/Th$ (for which we used 0.8 at $2\sigma$ throughout the paper), were propagated according to the specific method and tool applied. For the constant melt approaches (a) and (b), uncertainties were propagated using first-order Gaussian error propagation, whereas the other two constant melt approaches (c) and (d) followed the uncertainty propagation routines implemented in IsoplotR (Ludwig, 2003; Vermeesch, 2018).

In the constant $f_{U/Th}$ approach, using the code by Boehnke et al. (2016), the uncertainties of the zircon and melt activity ratios, along with the preset uncertainty on $f_{U/Th}$, were propagated through a parametric bootstrap resampling method (Efron, 1992). The code additionally accounts for the potential spread of the modeled melt composition around the equiline, for which we applied the suggested value of 0.3 at $2\sigma$ (Boehnke et al., 2016). ..."

"... we used the DQPB application by Pollard et al. (2023) to retrieve the $^{207}$Pb-corrected ages. The reported uncertainties of the ages follow the Monte Carlo uncertainty propagation (Sambridge et al., 2002). Initial disequilibrium was corrected by assuming a Th/U fractionation factor of 0.20 ± 0.04 between zircon and melt ..."

"... The two methods also propagate uncertainties differently. For the weighted mean, the uncertainties are calculated analytically using the standard formula for inverse-variance weighting (similar to Vermeesch, 2018). In the Bayesian method, uncertainties are derived from the distribution of eruption ages sampled via Markov Chain Monte Carlo (Metropolis et al., 1953). After discarding the initial burn-in samples, the mean and standard deviation of the remaining samples provide the eruption age estimate and its $1\sigma$ uncertainty (Keller et al., 2018). ..."

**RC:** Several studies have highlighted the importance of applying chemical abrasion treatment to zircon prior to LA-ICPMS U-Pb dating (e.g., Crowley et al., 2014; Von Quadt et al., 2014; McKanna et al., 2023; Donaghy et al., 2024). While Pb loss is unlikely to be a significant issue in <1 Ma zircons, the zircon standards used in this study are considerably older and likely to have experienced substantial radiation damage and associated Pb loss. Could the authors clarify whether chemical abrasion was applied to the standards, and if not, how potential Pb loss was addressed in the analytical protocol? This information would help readers assess the reliability of the standardization and the resulting age determinations.

**AR:** See comment above. Since our unknown samples are not annealed, we did not use annealed zircon standards.

**RC:** The Eruption Age Calculation Methods section (pg. 8) is currently difficult to follow due to its organization. Much of the confusion arises from switching back and forth between weighted means and Bayesian models. I recommend reorganizing this section so that you present the weighted mean and Bayesian model approaches in separate paragraphs. For each, begin by discussing the current status quo for that calculation type, including common pitfalls, and then describe the specific models you will test. Abbreviations should be used sparingly; for example, I suggest not using "WM." Also, since your weighted mean models are not proper nouns, they should be capitalized only as appropriate (e.g., "young" and "acceptable"). Additionally, please define the "I" in i-MSWD—I assume it stands for "iterative." For the i-MSWD approach, clarify how other ages are added to the population: how many older ages are included at each step? Providing these details will make the methodology much clearer for readers.

**AR:** Thanks for the comment. We removed the "WM" abbreviation, and we added clarity to the "i-MSWD" method. However, it was important to us to first introduce the status quo of both methods and compare their underlying assumptions, their similarities, and their strengths.

"... (2) The second approach is the so-called "i-MSWD" weighted mean (Popa et al., 2020), where the mean squared weighted deviation (MSWD) is calculated iteratively, hence the name, by progressively adding older ages to the population. At each step, one additional older analysis is included, and the youngest coherent population is identified based on changes in the iterative MSWD values. These changes are evaluated visually from a plot, where distinct jumps in the MSWD indicate the inclusion of older age populations (Popa et al., 2020). ..."

**RC:** All the figure captions could use addition detail. Whenever possible, a figure and its caption should be able to stand-alone to improve comprehension. Most of the figure captions lack mention of

**AR:** See comments below.

**RC:** All figure captions would benefit from additional detail. Ideally, each figure and its caption should be able to stand alone, providing enough information for readers to understand the figure without referring back to the main text. Since KPT is the only sample measured using the U-Th-Pb method, figure S1 should be moved from the supplementary materials into the main text to improve clarity and accessibility. For the U-Th isochron plot, the error ellipses appear to represent 1 sigma uncertainties, even though the caption states they are 2 sigma (in Fig S1). Please double-check all isochron plots to ensure the error ellipses accurately reflect the stated uncertainties (1 sigma vs. 2 sigma). I also note that most figures do not specify what the plotted uncertainties represent. Clearly stating whether uncertainties are 1 sigma, 2 sigma, or another value is essential for proper interpretation of the data.

**AR:** We have added more detail to the figure captions and clearly stated the uncertainty levels for all figures. Providing additional information on the specific model age approaches or eruption age estimation methods within each caption would, however, go beyond their intended scope.

We agree and thank you for pointing it out, as it is important to state whether 1 or 2 sigma is plotted. The uncertainties/error ellipses of Figure S1(b) are plotted in 2sigma as stated. All the data is available in the supplementary files. The resulting model ages have high uncertainties not because of high analytical uncertainties, but due to the exponential convergence towards secular equilibrium as illustrated by the red isochron lines.

**RC:** In the outlook section, you mention the asymmetric nature of your model ages and how they are not used in your models. It would be worthwhile to provide some text in the discussion about how this might bias your models and the eruption age estimates.

**AR:** We agree that the asymmetric nature of the model ages is an important consideration and have highlighted it in the outlook section. A detailed quantitative assessment of its influence on eruption age estimates would require a more complex mathematical treatment than what is feasible within the scope of this paper, but we recognize its relevance for future analyses.

Technical Corrections:

**RC:** Line 20: delete magmatic (repeated twice in sentence)

**AR:** The repetition has been removed. Thank you for pointing that out.

"... Such alteration frequently degrades minerals that would otherwise record magmatic conditions, ..."

**RC:** Line 27: delete an

**AR:** Done.

"... Since accurate chronology ..."

**RC:** Line 30: It is a stretch to say that U-Pb is widely used for >150 ka. I can only think of a few SIMS or LA-ICPMS 206Pb/238U zircon studies for samples <1 Ma. Add citations

**AR:** This was meant to illustrate the age range of the applicability of the overall dating tools. We adjusted the phrasing and added references to the age limits.

"... The most widely used geochronological methods for zircon are U–Pb dating, applicable to crystals older than ~100 ka (Sakata, 2018), and U–Th disequilibrium dating, which is suitable for ages younger than ~380 ka (Schmitt, 2011). ..."

**RC:** Line 41: You only list zircon (U-Th)/He for dating eruptions. How about 40Ar/39Ar? Throughout the paper you use crystallization but sometimes you use crystallisation. Be consistent.

**AR:** We thank the reviewer for this suggestion. We are focusing on zircon dating in this paper because zircon is a highly robust mineral that survives substantial alteration. Therefore, we decided not to discuss 40Ar/39Ar dating in detail to avoid diverting from the main focus.

Thanks for pointing that out. We changed all "crystallisation" to "crystallization".

**RC:** Line 46: This sentence is awkward, consider rephrasing. Replace "Instead of dating the eruption directly" with Alternatively. Should "through" actually be "from"?

**AR:** Thanks for the suggestion, we changed the phrase.

"... Alternatively, we can infer the eruption age from the latest zircon crystallization, ..."

**RC:** Line 48: Replace "Typical extended" with "Protracted". Consider using a different phrase from global. Global weighted mean and global isochron ages are vague terms so they need to be defined or use a different term.

**AR:** We replaced as suggested.

"... renders weighted mean and isochron ages derived from all analyses unreliable for eruption age estimation...."

Additionally, we defined the term "global" at its first use.

"... (d) through the global (hereafter referring to calculations encompassing all analyses from the sample of interest) isochron intercept ... "

**RC:** Line 49: Refs. Bachmann et al., (2007) is not the only group to make this observation. See the general comment above about reference issues throughout this paper.

**AR:** We cited Bachmann et al. (2007) because it was the first to specifically note that zircon ages reflect extended crystallization. We appreciate the comment and have added additional references to acknowledge other studies reporting similar observations:

"... Protracted zircon crystallization, often reflected in a broad spread of non-overlapping dates within a single sample (Bachmann et al., 2007; Schaltegger et al., 2015; Klein and Eddy, 2024), ..."

**RC:** Line 51: What are weighted ages of zircon subsets?

**AR:** We rephrased it for clarity.

"... including a weighted mean of a subset of zircon ages ..."

**RC:** Line 52: Include Keller et al., 2018 in the list of refs and check the order of refs.

**AR:** The reference was already included in the list. Thanks for the hint, we will make sure to go from oldest to newest reference within brackets.

**RC:** Line 55: What do you mean by low variability in uncertainties? I think you might be trying to say that there is no systematic change in 206Pb/238U age uncertainties with time but that is not necessarily true. Young zircons ($< 10$ Ma) typically have larger age uncertainties given there has been much time for radiogenic 206Pb ingrowth.

**AR:** We rephrased for clarity.

"... However, their study focused on ID-TIMS U-Pb data using synthetic datasets in which all ages were assigned the same uncertainty, a scenario more applicable to U-Pb than to U-Th datasets. In U-Th disequilibrium dating, uncertainties increase with age due to the method's inherent exponential convergence toward secular equilibrium (Schmitt, 2011). ..."

We agree that 206Pb/238U uncertainties can change systematically with age. However, for the timescales relevant to eruption age estimation from zircons within a single sample, this variation is typically minor and not systematic. By contrast, U–Th disequilibrium dating shows a pronounced increase in uncertainty with age within single samples due to the approach toward secular equilibrium within a relatively short timescale, making this effect critical to account for in U–Th datasets.

**RC:** Line 56: It is worth mentioning in the text that the large 230Th/238U age uncertainties are driven by the uncertainties in determining the isochron slope when it is close to unity.

**AR:** Thanks for the suggestion, but we will not implement this comment. We note that the uncertainty in determining the 230Th/238U isochron slope remains essentially constant; but rather, as stated, the propagated age uncertainty from the isochron increases because the system converges exponentially toward secular equilibrium.

**RC:** Line 60: Most of the body of work is focused on using 230Th/238U zircon crystallization ages to determine an eruption age. There isn't much discussion about quantifying eruption frequencies and timescales, consider rephrasing or add text in the discussion to this point.

**AR:** Thank you for this comment. We agree that the manuscript does not directly address the quantification of eruption frequencies and timescales. Our intention is to emphasize that zircon crystallization ages and eruption age estimates represent a fundamental basis for subsequently establishing eruption frequencies and timescales. We have rephrased the relevant section for clarity.

"... To improve the application of U–Th zircon dates for establishing a fundamental part of the geochronological framework, and thereby contributing to quantifying eruption frequencies and magmatic timescales,

we first evaluate ...”

**RC:** Line 62: Define LA-ICPMS at first use. By "methods," are you referring to 230Th/238U vs 206Pb/238U or age modeling?

**AR:** We clarified and rephrased.

"... we first evaluate the two opposing model age approaches (constant melt vs. constant U/Th fractionation), followed by an assessment of different eruption age estimation methods (weighted mean vs. Bayesian) for typical U–Th zircon datasets measured by laser ablation inductively coupled plasma mass spectrometry (LA-ICP-MS). ...”

**RC:** Line 64: Check citation order? Oldest first?

**AR:** We checked the order, and changed it everywhere to oldest first.

**RC:** Line 65: Do age determination techniques refer to analytical or modeling?

**AR:** Thanks for the comment, we rephrased.

"... we assess the reliability of different ways to calculate U–Th model ages. ...”

**RC:** Line 67: Define typical LA-ICPMS uncertainties (at 1 vs 2 sigma?).

**AR:** As this is not a straightforward number, we do not want to add the values here, but we agree that they should be mentioned within the manuscript. We added them in the methodology part (Line 175), where we are explaining the structure of our synthetic dataset.

"... While we can account for the age dependency by assigning higher uncertainties to the older ages, we also allow the uncertainty to spread around a mean value to account for variable U/Th ratios. For illustration, this corresponds to $1\sigma$ uncertainties of approximately 4.5–10.5 ka at 20 ka and 11–25 ka at 100 ka. ...”

**RC:** Line 72: Do different methods refer to analytical methods? See comments above about "developed new dating routine”

**AR:** Thanks for pointing out that this was not sufficiently clear. We rephrased accordingly.

"... To assess the different approaches of calculating the U-Th crystallization ages (constant melt or constant $f_{U/Th}$), we developed an optimized LA-ICP-MS strategy to simultaneously measure U-Th-Pb (similar to Ito, 2014, 2024), and to compare the differently calculated U-Th ages with the U-Pb ages of the same ablation volume. ...”

**RC:** Line 76: I don't really understand what you are trying to say in the sentence "Given that. . .” Is U-Pb dating better understood than U-Th dating? Probably also depends on the time period you are talking about. How is it better understood? Rephrase sentence. Also check refs.

**AR:** Thanks for pointing out that it was not clear. We rephrased this part.

"... Although young U–Pb ages require corrections for initial U-Th disequilibrium and common lead, these corrections are well understood, making the overall age calculation relatively straightforward (Sakata et al., 2017; Pollard et al., 2023). In contrast, U–Th model ages depend on the isotopic composition of the melt in equilibrium with the zircon, for which there is currently no consensus on how to estimate it. Consequently, U–Pb ages provide a useful benchmark for evaluating the performance of different U-Th model age approaches (Sakata et al., 2017; Pollard et al., 2023). ...”

**RC:** Lines 78-80: Sentence " Using addition” is awkward. Rephrase. You alternate between analyze and analyse. Be consistent.

**AR:** We rephrased and changed all "analyse" to "analyze".

"... We then validate these approaches by applying the best model age calculation methods to U-Th datasets for samples with well-constrained eruption ages. ...”

**RC:** Line 83 (and elsewhere): use 40Ar/39Ar. Also this sentence is a run-on. Rephrase.

**AR:** Thanks for pointing this out. We changed it throughout the manuscript and rephrased the sentence.

"... Thórsmörk is an ignimbrite that was most recently dated by $^{40}Ar/^{39}Ar$ measurements of glassy fiamme to an age of 56.14 ± 0.44 ka ($2\sigma$) by Groen and Storey (2022), which overlaps with other independently published ages (Svensson et al., 2008; Guillou et al., 2019; Moles et al., 2019). ..."

**RC:** Line 91: Change to beginning of sentence to: The LA-ICPMS measurements were conducted at ETH Zurich using 193 nm. . . .

**AR:** We applied your suggestion.

"The LA-ICP-MS measurements were conducted at ETH Zurich using a 193 nm Resonetics Resolution 155 LR excimer laser ablation system coupled to a Thermo Element XR sector field mass spectrometer."

**RC:** Line 92: change "to a" to and

**AR:** We rephrased the sentence, see comment above.

**RC:** Line 94: Did you calculate a ACF at the start of each run? Or what is a common ACF value?

**AR:** Yes, the ACF factor is determined before every session based on Uranium. The absolute value depends on the age of the detector.

"... were measured with an analog detection mode and a common analog to puls-counting equivalent factor (ACF) determined based on uranium before each session during calibration,..."

**RC:** Line 98: What is a zircon blank?

**AR:** We specified in the text.

"... and a synthetic zircon, free of any U and Th, ..."

**RC:** Lines 104-106: What do you mean by "independent of the measuring method"?

**AR:** We specified in the text.

"... Independent of the LA-ICP-MS measuring routine (U-Th-Pb or U-Th), the processing of the U-Th data..."

**RC:** Paragraph starting at Line 112: I think it would be better to use the term partition coefficient than fractionation factor. This paragraph might be easier to read if it was summarized in a table. Factionation is misspelled (line 125).

**AR:** The term "U/Th fractionation factor" is used intentionally because it represents the ratio of the U and Th partition coefficients between zircon and melt, rather than the absolute partition coefficients themselves, as we are interested in their relative fractionation.

We have corrected the spelling.

**RC:** Line 145: List which primary reference zircon you are referring to? 91500? Which statistically robust methods do you suggest? Expand on this statement. (What are physically robust methods?)

**AR:** According to Table 1, we used 91500 as our primary reference zircon, but we added the information to the text again.

"... using the corresponding time slices of primary reference zircon 91500 ..."

We are citing a new method that is still being developed. We added a Conference abstract in the references and expanded the explanation:

"... by adopting statistically more robust methods (Vermeesch, 2022) or physics-based algorithms that fit models to the data based on the fundamental age equation, rather than following the conventional

approach of fitting the data to a model, thereby avoiding the biases of heuristic methods (Vermeesch and Glorie, 2025; Vermeesch, 2025). ..."

**RC:** Line 147-148: "To finally..." is an awkward sentence, rephrase. Also define DQPB abbreviation. Change to "Initial 230Th disequilibrium" in the next sentence. You alternate between DTh/U and fU/Th. Be consistent. Why did you select 0.2 for DTh/U? Based on glass measurements that you collected (mean value? or n of 1?)? How much do your 206Pb/238U change if you vary the DTh/U?

**AR:** We rephrased the start of the sentence. DQPB is the name of the application, not an abbreviation. We cited the corresponding paper.

"... To calculate the U-Pb ages of the young volcanic zircon to be later compared with the U-Th ages for the same ablation volumes, we used the DQPB application by ..."

The DQPB application specifically asks for the $f_{Th/U}$ rather than the inverse $f_{U/Th}$, which is why we used this. As the uncertainty of the inverse would be asymmetric, for correctness, we keep the values we actually applied. But we specified why we selected 0.2. We mentioned in the results how much the change in age would be, if we used 0.25 instead. We moved this now to the methods.

"... Initial disequilibrium was corrected by assuming a Th/U fractionation factor of $0.20 \pm 0.04$ between zircon and melt, which lies between the published values of 0.25 and 0.18 for the KPT (Guillong et al., 2014). Using a Th/U fractionation factor of 0.25 to correct for initial Th disequilibrium would yield ages roughly 5 ky younger. ..."

**RC:** Paragraph starting at line 153: Quantify your parameters of your synthetic dataset. How many zircons and duration of zircon crystallization (did you do a literature review to pick a duration)? What is your analytical uncertainty?

**AR:** We added the parameters and clarified in the text why we chose these periods:

"... For this study, we specifically simulated two different zircon crystallization periods: 0–40 ka (eruption at 0 ka, $\Delta t = 40$ ky) and 55-170 ka (eruption at 55ka, $\Delta t = 115$ ky), to cover a range of lower and higher analytical uncertainties as well as different crystallization durations. For $N_{zircon}$, we choose values between 10 and 150, which are reasonable for typical LA-ICP-MS datasets. ..."

As mentioned above, we added examples of assumed analytical uncertainty.

"... While we can account for the age dependency by assigning higher uncertainties to the older ages, we also allow the uncertainty to spread around a mean value to account for variable U/Th ratios. For illustration, this corresponds to $1\sigma$ uncertainties of approximately 4.5–10.5 ka at 20 ka and 11–25 ka at 100 ka. ..."

**RC:** Line 157: Remove "First," redundant with (1). Change to: representing crystallization from the time of zircon saturation until eruption. Should you reference Fig. 2C somewhere in this paragraph?

**AR:** Thanks for this suggestion. We rephrased the sentence. Fig. 2d corresponds to the prior distributions for the likelihood-based Bayesian eruption age estimate method, not to how we generate the synthetic datasets.

"... The model must randomly sample zircon dates from an underlying distribution representing zircon crystallization from zircon saturation until eruption. ..."

**RC:** Lines 168-169: Change to constant zircon production to continuous zircon crystallization. Define the sampled time interval? What is a crystallization bloom? Like an algae bloom?

**AR:** Thanks for the pointers. We rephrased and clarified.

"... suggesting relatively constant zircon crystallization throughout the period in which the system is zircon saturated. This can be translated into representing many non-resolvable periods of intensified zircon crystallization ..."

**RC:** Line 208: You previously mention using 0.2 (line 149).

**AR:** This was meant to illustrate how much the age changes if, instead of 0.2, we had used 0.25. As mentioned above, we moved this sentence into the methods.

**RC:** Line 212: Can you quantify "higher uncertainties"? Also, rephrase "For the method" sentence. It is a run-on sentence.

**AR:** Good point. We clarified and rephrased.

"... The U-Th ages based on the different approaches and approximations show a rough spread between 140-350 ka, with on average four times higher uncertainties compared to the U-Pb ages. For the constant melt approach, ..."

**RC:** Line 214: Which model age assumptions are you referring to here? Providing an actual value or range is more useful than descriptors like "infinitely high uncertainties"?

**AR:** We literally mean infinitely high uncertainties because of secular equilibrium. We rephrased for clarity.

"... For the constant melt approach, which uses the measured $^{230}$Th/$^{232}$Th and $^{238}$U/$^{232}$Th activities in the groundmass glass, fewer model ages were resolved because many data points approached secular equilibrium and therefore produced infinitely large uncertainties. For the different U-Th model age approaches, individual U-Th and U-Pb ages are generally in good agreement within $2\sigma$ uncertainty. ..."

**RC:** Line 216: Unclear if "their lower reported uncertainties" refers to U-Pb or U-Th.

**AR:** Thanks for pointing that out. We rephrased for clarity.

"... However, IsoplotR U–Th model ages using the measured U/Th ratio in the groundmass glass report lower uncertainties (on average half those of other U–Th ages), resulting in poorer agreement with the corresponding U–Pb ages compared to other U–Th model age approaches. ..."

**RC:** Line 223: Awkward sentence. Rephrase.

**AR:** We rephrased.

"... In contrast, a constant melt composition suggests that the magma chamber was homogeneous during zircon crystallization, such that variations in the U/Th ratio were negligible ..."

**RC:** Line 226: What does "dilution of the signal" mean? Be very specific with you are trying to say.

**AR:** Thanks for pointing out, that it was not clear. We rephrased to clarify.

"... Subsequently, any U/Th variations in the zircon would either result from variable U and Th partitioning behaviour or from mineral inclusions within the crystal, effectively mixing the zircon isotopic compositions with that of the inclusion. ..."

**RC:** Line 231: Here you say that a constant fractionation factor is incorrect though at the beginning of the paragraph you say that assuming a constant factor permits spatial and temporal heterogeneity in the melt. I don't follow the contradiction. Expand on both arguments and in which situations these would matter.

**AR:** As stated in the text, this refers to the measured isotopic composition consisting of a mixture of zircon and apatite. We rephrased for clarity.

"... In such cases, assuming a fixed fractionation factor is incorrect, as these inclusions often have different partitioning behaviour than zircon (Blundy and Wood), and is therefore more appropriately dealt with in the constant melt composition method. ..."

**RC:** Line 232: What do you mean by boundary layers of other mineral phases?

**AR:** The boundary layer is a well-established concept in petrology: e.g. "The boundary layer is the region adjacent to the interfaces, characterized by concentration gradients in the melt owing to dissolution or growth of [crystallizing phases].", as defined by Bindeman and Melnik (2016).

This term has been used in the literature for decades, which is why we do not further elaborate on this term in the text. We specified in the text what we mean, but we added a reference. And in the hopes for clarity, we rephrased slightly.

"... However, the constant melt approach may be questioned when many accessory phases crystallize in the boundary layers of other minerals (Bacon, 1989; Bindeman and Melnik, 2016). In these regions, the melt can become locally enriched in incompatible elements such as U and Th, potentially fractionating them and altering the composition of the melt from which the zircon grows. ..."

**RC:** Line 237: Include more refs

**AR:** We added two more references.

"... Melt can be influenced by pre-eruptive mixing processes (e.g. Nakamura, 1995; Troll et al., 2004), and zircon datasets show large variations in trace elements (e.g. Troch et al., 2018; Bell and Kirkpatrick, 2021; Castellanos-Melendez et al., 2024), undermining the assumption of a constant melt composition. ..."

**RC:** Paragraph stating at line 238: Paragraph could use some restructuring. I'm not exactly sure the point you are trying to make in this paragraph. Change "might be even more complicated" to "is complex". Define CL. It feels like you are rushing to make a series of statements, without providing some context for each statement or a way to tie it altogether into a coherent paragraph.

**AR:** Thanks for the comment. We have edited the paragraph to give more reasoning to the statements.

"... Zircon crystallization is further complicated by undercooling and disequilibrium growth. Gillespie et al. (2025) showed that undercooling can lead to dendritic growth in zircon, where apparent cathodoluminescence oscillatory growth rims exhibit variable U/Th compositions. During this process, the crystal corners grow rapidly and incorporate lower U/Th values, whereas slower infilling into the planar structures produces higher U/Th values. This disequilibrium growth explains U/Th variability within zircon without invoking changes in melt composition or inclusions, highlighting that bulk partition coefficients may not accurately describe zircon grown out of equilibrium (Gillespie et al., 2025). Similarly, sector zoning represents a form of homogenous disequilibrium within zircon (Watson and Liang, 1995) and can influence trace element partitioning, even in a chemically homogeneous melt (Burnham and Berry, 2012; Burnham, 2020). Such zoning reflects variations in crystallographic orientation that can locally modify partition coefficients and result in heterogeneous trace element distributions within a single crystal (Burnham and Berry, 2012). Beyond crystal-scale effects, melt evolution also seems to systematically affect the U/Th fractionation, as the $fU/Th$ decreases with increasing silica in the whole rock Kirkland et al. (2015). The oxygen fugacity of the system further influences U/Th partitioning because U incorporation depends on its oxidation state ($U^{4+}$ versus $U^{6+}$), such that $fU/Th$ decreases under more oxidizing conditions (Burnham and Berry, 2012). Together, these factors highlight that U–Th partitioning is sensitive to both crystal-scale and system-scale processes, casting doubt on the assumption of a constant fractionation factor between zircon and melt. This underscores the complexity of the zircon–melt system and highlights that both model age approaches (constant melt versus constant $f_{U/Th}$) rely on end-member assumptions. ..."

**RC:** Line 256: Awkward sentence, rephrase. What do you mean by measure glass accurately? Analytical precision? Or would it be worthwhile measuring different shards of glass?

**AR:** We wanted to point out that 238U/232Th can more accurately be determined, due to higher concentrations and therefore better precision, as well as less corrections needed during data processing. Typically, we measure 10-15 points, spread over 5-8 shards. But we understand your point. We rephrased.

"... Due to higher concentrations in the melt, measuring $^{238}U/^{232}Th$ in the groundmass glass can be done more accurately and with higher precision. Therefore, a more conservative approach assumes the melt is in or close to secular equilibrium, supported by Boehnke et al. (2016), using the measured $^{238}U/^{232}Th$ in the groundmass glass. This approach ..."

**RC:** Line 262: Reference a figure. Figure 3?

**AR:** We added the reference to a figure.

**RC:** Line 268: Rephase "This is potentially. . . " sentence.

**AR:** We rephrased the sentence.

"... The reduced sensitivity of older zircon to variations in melt composition or fractionation factor may account for the lack of significant differences observed in the KPT crystals when comparing model ages derived using different assumptions of constant melt composition or constant fractionation factor. ..."

**RC:** Line 276: Define enhanced 232Th counts (counts > XX)

**AR:** Good point, thanks, we included the number and the method of identification.

"... About 9% of the data points show strongly elevated $^{232}$Th concentrations of >485 ppm Th identified as statistical outliers using the 1.5 times interquartile range (IQR) criterion, indicating possible contributions from apatite inclusions. ..."

**RC:** Line 279: Awkward sentence, rephrase. Are there inclusions in the glass? "Fall well above" could be rephrased to older than

**AR:** We rephrased it. The groundmass glass is fresh and homogeneous, but contains many microlites that grew post-eruption during cooling of the lava. As the 14 groundmass glass measurements are generally in good agreement, we do not expect any small inclusions in the glass to divert the melt composition homogeneously and along the secular equilibrium line.

"... Even the youngest calculated model ages using the measured groundmass glass either as a constant melt anchor point or to estimate the fractionation factor are significantly older than the eruption age ..."

**RC:** Line 286: Change to zircon-bearing crystal mush. I think you should circle back to your observation above that the youngest zircon ages are older than the independently observed eruption based on your arguments here. It is not uncommon for the crystal mush (including zircon crystals) to be scorned from deeper parts of the magmatic system just prior to eruption, which could explain why you don't see overlapping ages between zircon and eruption ages (i.e., the recharge melt didn't reach zircon saturation conditions). I think you could expand and emphasize that point.

**AR:** We changed to "zircon-bearing crystal mush".

We agree that many zircons could potentially be sourced from different parts of the magmatic system and remobilized just before eruption. But importantly, the mixing with recharge happened immediately pre- and syn-eruptive (Nakamura, 1995). Therefore, this process does not explain the observed gap between zircon ages and eruption ages. There is no reason to assume that zircon crystallization stopped in the more crystal-rich, mechanically locked portions of the crystal mush, nor that only older zircons were selectively entrained. Instead, this observation supports the idea that the erupted melt may not have been in equilibrium with the crystallizing zircon population, particularly if zircons were scavenged from different parts of the crystal mush.

**RC:** Line 293: Change intercept to y-intercept.

**AR:** Thanks for the comment. It is important that the young isochron intercept does not refer to the y-axis intercept, but to the intercept with the secular equilibrium line. Therefore, we introduce the term $y_0$.

**RC:** Line 310: Was data with high Th concentrations excluded from the figures and age modeling?

**AR:** Good point, we will make that clear in the text and the figure captions.

In model age calculations assuming a constant melt, the high Th data were used as there is no reason to think that the zircon and the inclusion where not derived from the same melt, however in the model age calculations assuming a constant fractionation factor, they were excluded, as the fractionation factor is meant to represent pure zircon-melt fractionation. In this regard, for the measurements recording high Th, the model ages are only plotted when derived from the constant melt approach.

"... Apatite inclusions are evident in about 6% of the isotopic signals based on enhanced Th concentrations

($>425$ ppm for LH, $>155$ ppm for TH) identified as statistical outliers. For those measurements, only the model ages calculated with the constant melt approach were calculated. The model ages that use the measured $(^{238}U)/(^{232}Th)$ in the groundmass glass ..."

Similarly, following line 276:

"... indicating possible contributions from apatite inclusions. These measurements were therefore excluded from the model age calculations based on the constant $f_{U/Th}$ approach, as it assumes pure zircon–melt fractionation. ..."

**Figures/Tables**

**RC:** Figure 1: Use "constant melt composition" as opposed to constant glass composition. The glass composition is a proxy for the melt composition. Are uncertainties 1 or 2 sigma?

**AR:** Thanks for realizing that was an oversight. We changed it to "constant melt composition". And we added a comment clarifying that the uncertainties represent 2 sigma.

**RC:** Figure 2: Are you showing 1 or 2 sigma uncertainties? (a) Add 230Th/238U to y-axis. What is a "true eruption age" in a synthetic dataset? Also mention what the typical LA-ICPMS uncertainties that were used (e.g., 10%, 2 sigma). (b) Define the different weighted mean models that are represented by the three colored symbols in caption. (c) There is no explanation for what the different lines (solid and dashed) mean. What are tested zircon age subsets? (d) Provide a more detailed explanation of both axes in the caption. I think x-axis refers to time though I'm not sure if the dash means subtraction or from.

**AR:** We represent 1 sigma uncertainty for this synthetic dataset, since this is the uncertainty we have to use as an input into a weighted mean calculation or Bayesian calculation. We specified in the caption. The true eruption age is a preset value in the routine to generate the synthetic datasets. We rephrased the caption. We added more details to the caption as well.

"... Example of a synthetic $^{238}U$–$^{230}Th$ LA-ICP-MS zircon crystallization age dataset and comparison of eruption age estimation methods. (a) Synthetic ranked dataset of 50 zircon $^{238}U$–$^{230}Th$ crystallization ages $(1\sigma)$, simulating a natural volcanic system with ages skewed toward the eruption (Nathwani et al., 2025). Ages range from a preset eruption age of 55 ka to a zircon saturation age of 170 ka and include typical LA-ICP-MS uncertainties. (b) Iteratively calculated weighted mean ages $(1\sigma)$. The colors correspond to the tested subsets of zircon ages for the weighted mean method to estimate the eruption age. (c) Corresponding iterative MSWD values, shown alongside the tested subsets of zircon ages (solid lines) used to estimate the eruption age via the weighted mean approach. The "youngest 10%" weighted mean is calculated from the youngest 10% of zircon ages. The "i-MSWD" weighted mean corresponds to a subset of young zircon ages before, according to Popa et al. (2020), a visual increase in the iteratively calculated MSWD indicates the addition of older ages. The "acceptable MSWD" weighted mean follows a criterion as a function of the number of datapoints (N, dashed line) at which it remains possible that the zircon ages represent an isochron age (Wendt and Carl, 1991). (d) Relative zircon crystallization distributions between zircon saturation (1) and eruption (0). The distributions represent three different scenarios of normalized crystallized zircon mass as a function of time and are tested as prior distributions for the likelihood-based Bayesian eruption age model (Keller et al., 2018). While the truncated normal and uniform distributions are fixed, the bootstrapped distribution is obtained through the kernel density function of the zircon age dataset. ..."

**RC:** Figure 3: The error bars and light gray circles are nearly impossible to see if you print them. Is it possible to make use a different color or shape for the light gray symbols. Could you also make these figures bigger? The x-axis for (a-d) is misleading. It also looks like the they are the partition coefficients. Either add the x-axis labels back in or increase the space between the two rows and mention in the caption that they all have the same x-axis. I'm not sure what the difference between the two RMS-Z means (the values not in parentheses). You mention normalized age difference, what are you normalizing to?

**AR:** We understand and agree with your point. We increased the darkness of the light grey circles

and the errorbars and turned the light grey circles into squares. We increased the size of the figure to cover the width of the page. We added the x-labels to all panels. The normalization is in regard to the combined uncertainties. We added this detail to the caption.

"... Each panel shows the root mean square of the age differences normalized by the combined uncertainties of both datasets (RMS-Z) for the overlapping and total age datasets, along with the number of points considered (overlapping | total). RMS-Z values ≥1 indicate decreasing agreement between the U–Th and U–Pb age estimates. ..."

**RC:** Figure 4: How is the "true eruption age" determined? Mention in caption. Add error ellipses are 1 sigma uncertainty. There are some ages younger than the visual near-zero isochron line. What criteria did you use to make your near-zero isochron line? There is no mention in the caption what the colored symbol.

**AR:** The eruption happened 1991-1995 and has been observed and occurred in living memory. We consider this fact to be common knowledge, and therefore we do not add a reference here.

Good point, the uncertainties are shown in 2 sigma, we added this to the caption.

The visual young isochron line was drawn by making sure that the youngest age is at least within 2 sigma uncertainty of the observed eruption age of ∼0 ka.

We added the point about the colors in the caption.

"... Validation of U–Th zircon age determinations using the Heisei sample from the historically observed 1991–1995 Unzen eruption (all uncertainties correspond to $2\sigma$). (a) U–Th evolution plot showing measured zircon and groundmass glass activity ratios, the young isochron intercept with the secular equilibrium line evaluated using IsoplotR (Vermeesch, 2018), and the visually identified young near-zero age isochron. The latter was drawn to ensure that the youngest age is within $2\sigma$ of the known near-zero eruption age. The discrepancy between the measured groundmass glass and the young isochron intercept indicates that the groundmass glass does not represent the melt from which the zircon crystallized. (b) Zircon crystallization ages using different approaches are plotted in grey tones, while eruption age estimates for the Heisei sample, calculated using different methods, are shown in color. The marker shape corresponds to the different model age datasets. ..."

**RC:** Figure 5: I think the dashed lines should be removed. I'm not sure they add much to the figure. Just FYI, Barboni et al, 2016 also has U-Th data for the Belfond Dome too. What are the red lines in the inset (c).

**AR:** We specifically decided to use the dataset by Schmitt et al. (2010), since it was also used as a case for proposing using the constant fractionation factors to estimate U-Th ages by Boehnke et al. (2016).

The red lines correspond to the (U-Th)/He eruption age, as in the main panel. We specified.

**RC:** Figures 6: define colored symbols (MSWD and bayesian models). The three symbols on the right side of the (b) are very small and hard to see. Are there no constant melt circles in (b). Define uncertainties.

**AR:** Thank you for the suggestions. We extended the figure caption, increased the size of the stars, and changed the plotting order of the symbols. The circles were hidden behind the squares because the two model age approaches resulted in similar crystallization ages.

"... Zircon crystallization and eruption age estimates for two validation samples from the Torfajökull volcanic system, Iceland (all uncertainties correspond to $2\sigma$). (a–b) Zircon crystallization ages (grey symbols) and eruption age estimates (colored symbols) for two samples from Torfajökull, with the marker shape corresponding to the different model age datasets. The first three estimates represent weighted mean ages calculated from different subsets of zircon ages, whereas the remaining three are Bayesian eruption estimates based on different prior distributions. Measurements of enhanced Th, implying apatite inclusions were ablated along with the zircon, were only used to calculate model ages with the constant melt approach. Shown alongside are suggested eruption ages for Laugahraun based on tephrochronology

(Larsen, 1984) and the most recently published $^{40}$Ar/$^{39}$Ar age of glassy fiamme from the Thórsmörk ignimbrite (Groen and Storey, 2022). The stars on the right of (b) correspond to independent eruption age estimates for Thórsmörk: (1) Moles et al. (2019), (2) Svensson et al. (2008), (3) Guillou et al. (2019). ...”

**RC:** Table 1: What is the difference between the U-Th-Pb zircon and U-Th zircon columns? I thought all your zircon analyses represent simultaneous U-Pb and U-Th analysis? (It was not clear to me until I looked at your raw data tables that U-Th-Pb measurements were only conducted on The KPT samples. This should be clearly stated in the methods section.)

**AR:** Thank you for pointing that out. We agree it is important to state it more clearly. We adjusted the table to add a row of "Measured samples" for the given measurement routines, and we pointed out the differences in the table caption.

[revised manuscript text omitted]